**Title**
**Enhanced Stratospheric Water Vapor over the Summertime**
**Continental United States and the Role of Overshooting Convection**
Robert L. Herman[1], Eric A. Ray[2], Karen H. Rosenlof[2], Kristopher M. Bedka[3], Michael J.
Schwartz[1], William G. Read[1], Robert F. Troy[1], Keith Chin[1], Lance E. Christensen[1], Dejian Fu[1],
Robert A. Stachnik[1], T. Paul Bui[4], Jonathan M. Dean-Day[5]
[1]Jet Propulsion Laboratory, California Institute of Technology, Pasadena, California, USA.
[2]National Oceanic and Atmospheric Administration (NOAA) Earth System Research Laboratory (ESRL) Chemical
Sciences Division, Boulder, Colorado, USA.
[3]NASA Langley Research Center, Hampton, Virginia, USA.
[4]NASA Ames Research Center, Moffett Field, California, USA.
[5]Bay Area Environmental Research Institute, Sonoma, California, USA.
*Correspondence to:* R. L. Herman (Robert.L.Herman@jpl.nasa.gov)

**Abstract**

The NASA ER-2 aircraft sampled the lower stratosphere over North America during the NASA Studies of Emissions and Atmospheric Composition, Clouds and Climate Coupling by Regional Surveys (SEAC[4]RS) field mission. This study reports observations of convectively-influenced air parcels with enhanced water vapor in the overworld stratosphere over the summertime continental United States, and investigates in detail three case studies. Water vapor mixing ratios greater than 10 ppmv, much higher than the background 4 to 6 ppmv of the overworld stratosphere, were measured by the JPL Laser Hygrometer (JLH Mark2) at altitudes between 16.0 and 17.5 km (potential temperatures of approximately 380 K to 410 K). Overshooting cloud tops (OT) are identified from a SEAC[4]RS OT detection product based on satellite infrared window channel brightness temperature gradients. Through trajectory analysis, we make the connection between these *in situ* water measurements and OT. Back trajectory analysis ties enhanced water to OT one to seven days prior to the intercept by the aircraft. The trajectory paths are dominated by the North American Monsoon (NAM) anticyclonic circulation. This connection suggests that ice is convectively transported to the overworld stratosphere in OT events and subsequently sublimated; such events may irreversibly enhance stratospheric water vapor in the summer over Mexico and the United States. Regional context is provided by water observations from the Aura Microwave Limb Sounder (MLS).

**Keywords**

Convection, overshoot, atmospheric water, stratosphere-troposphere exchange

## 1. Introduction

Water plays a predominant role in the radiative balance of the Earth's atmosphere, both in the gas phase as the Earth's primary greenhouse gas and in condensed phases in cloud and aerosol. Despite its low abundance, upper tropospheric and lower stratospheric (UTLS) water vapor is critically important in controlling outgoing long-wave radiation, and quantifying UTLS water vapor and its controlling processes is critical for climate characterization and prediction. Climate models are sensitive to changes in stratospheric water (Shindell, 2001) and clouds (Boucher et al., 2013). Increases in UTLS water are associated with warming at the surface on the decadal scale (Solomon et al., 2010). As the dominant source of hydroxyl radicals, UTLS water also plays an important role in control of UTLS ozone (Shindell, 2001; Kirk-Davidoff et al., 1999).

The overworld stratosphere, the altitude region with potential temperature $\theta$ greater than 380 K (Holton et al. 1995), is extremely dry, with typical mixing ratios of 3—6 parts per million by volume (ppmv). The importance of low temperatures at the tropical tropopause acting as a "cold trap" to prevent tropospheric water from entering the stratosphere has been recognized since Brewer (1949). Tropospheric air slowly ascends through the tropical tropopause layer (TTL) as part of the hemispheric-scale Brewer-Dobson circulation. In the TTL, air passes through extremely cold regions where water vapor condenses in situ to form cirrus ice, and then the cirrus slowly falls due to sedimentation (e.g., Jensen et al., 2013). Additional condensation and sedimentation are thought to be associated with convection and large-scale waves (e.g., Voemel et al., 2002). The amount of water that enters the stratosphere is largely a function of the coldest temperature a parcel trajectory encounters. This typically occurs in the tropics, and the coldest temperature is typically near the tropical tropopause. The saturation mixing ratio at the cold point tropopause thereby sets the entry value of water vapor.

In contrast to water entry into the overworld stratosphere, water transport from the troposphere to the lowermost stratosphere (350 K < $\theta$ < 380 K over summer CONUS) may occur through several different pathways. Poleward of the subtropical jet, water may be transported into the lowermost stratosphere through isentropic troposphere-stratosphere exchange (Holton et al., 1995) or through convective overshoot of the local tropopause (Dessler et al., 2007; Hanisco et al., 2007). Isentropic transport from the tropics is the dominant pathway for water into the lowermost stratosphere, with evidence from the seasonal cycle of lower stratospheric water (e.g., Flury et al., 2013). How important sublimation of ice from convective overshoot is for hydrating the stratosphere is a topic of ongoing debate (e.g., Randel et al., 2015; Wang, 2003). Case studies have reported extreme events in which ice is transported to the overworld stratosphere and subsequently sublimates, but the amount of ice that is irreversibly injected into the stratosphere is poorly known. Airborne measurements have demonstrated that convective injection occurs both in the tropics (Webster and Heymsfield, 2003; Corti et al., 2008; Sayres et al., 2010; Sargent et al., 2014) and at mid-latitudes (Hanisco et al., 2007; Anderson et al., 2012). Ice injected directly into the stratosphere is unaffected by the cold trap in the vicinity of the tropopause (Ravishankara, 2012).

The subject of this paper is the role of convective overshooting tops in enhancing stratospheric water. Paraphrasing Bedka et al. (2010), a convective overshooting top (OT) is a protrusion above a cumulonimbus anvil due to strong

updrafts above the equilibrium level. Early observations of OT include photographs of OT in the stratosphere from a
U-2 aircraft (Roach 1967). Recent observations of elevated water mixing ratios in the summer overworld
stratosphere by aircraft (Anderson et al., 2012) and the Aura Microwave Limb Sounder (MLS) (Schwartz et al.,
2013) suggest that ice injection into the overworld stratosphere by OT, while rare, occurs in three predominant
regions during the summer season.  These three regions are the Asian Monsoon region, the South American
continent, and – the focus of this study - the North American Monsoon (NAM) region (Schwartz et al., 2013).

The NASA ER-2 aircraft sampled the summer stratospheric NAM region during the NASA Studies of Emissions
and Atmospheric Composition, Clouds and Climate Coupling by Regional Surveys (SEAC[4]RS) field mission (Toon
et al., 2016). One of the primary goals of this multi-aircraft mission was to address the question: do deep convective
cloud systems locally inject water vapor and other chemicals into the overworld stratosphere over the continental
United States (CONUS)? It is challenging for space- and ground-based techniques to detect enhanced water vapor
injected into the stratosphere by OT. Satellite measurements are limited by their horizontal and vertical resolution in
detecting fine-scale three-dimensional variations in water vapor, while ground-based measurements are confined to
sampling at fixed locations. In contrast, airborne *in situ* stratospheric measurements of water have an advantage
because the aircraft can be routed to a specific location, altitude, date and time. Modelers can predict whether air
parcels are likely to have convective influence, and aircraft flight paths are planned to intercept those air parcels.
The purpose of this paper is to report three new case studies of enhanced water vapor in the overworld stratosphere
during the NASA SEAC[4]RS field mission, and to connect these observations to deep convective OT over the North
American continent.

**2. Observations**
**2.1 Aircraft**
The airborne *in situ* water vapor measurements reported here are from the Jet Propulsion Laboratory Laser
Hygrometer Mark2 (JLH Mark2), a tunable laser spectrometer with an open-path cell external to the aircraft
fuselage (May, 1998). Water vapor is reported at 1 Hz (10% accuracy), although the time response of the open-path
cell is much faster than this because the instrument is sampling the free-stream airflow. This instrument has a
redesigned optomechanical structure for greater optical stability, and was first flown in this configuration on the
NASA ER-2 high-altitude aircraft during the SEAC[4]RS field mission. Pressure and temperature, provided by the
Meteorological Measurement System (MMS) (Scott et al., 1990), are used in the data processing to calculate water
vapor mixing ratios from spectra, as described in May (1998).

During SEAC[4]RS, nine aircraft flights targeted air parcels with recent convective influence (see Table 3 of Toon et
al., 2016). Figure 1 shows the combined vertical profiles of JLH Mark2 water vapor from all 23 SEAC[4]RS flights.
Outliers with high water vapor mixing ratios are the focus on this study. Enhanced water vapor was measured on
eleven flights (Table 1). Here we define 'enhanced water vapor' as mixing ratios greater than two standard
deviations above the mean in situ measurement. For the overworld stratosphere in all 23 SEAC[4]RS flights, mean
$H_2O$ for is 6.7±1.5 ppmv at 380-400 K (Figure 2), and 5.0±0.8 ppmv at 400-420 K (Figure 3). Thus the threshold for
enhanced water vapor is 9.7 ppmv at 380-400 K, and 6.6 ppmv at 400-420 K. The majority of measurements have
background water mixing ratios characteristic of the overworld stratosphere, 4 to 6 ppmv. In the overworld
stratosphere (potential temperature greater than 380 K), Figure 1 shows enhanced water vapor at potential
temperatures up to approximately 410 K (17.5 km altitude). We define the 'enhanced water region' as the layer of
the overworld stratosphere where these events have been observed, 380-410 K potential temperature corresponding
to 16-17.5 km altitude. Enhanced water vapor measured *in situ* by both the JLH Mark2 instrument (Figure 1) and the
Harvard Water Vapor instrument (J. B. Smith, pers. comm.) on the NASA ER-2 aircraft indicated that the aircraft
intercepted convectively-influenced air. Other tracers measured on the aircraft did not change significantly in these
plumes. For the SEAC[4]RS flights, the agreement between these two water vapor instruments is within +/-10% for
stratospheric water. This is consistent with the AquaVIT laboratory intercomparison (Fahey et al., 2014). The largest
enhancements were observed on three flights that are described in detail in Sect. 4.

**2.2 Aura MLS**
Aura MLS measures ~3500 profiles each day of water vapor and other atmospheric species (Livesey et al., 2016).
While the aircraft samples *in situ* water in a thin trajectory through the atmosphere, Aura MLS provides a larger
scale context. Expanding on the analysis of Schwartz et al. (2013), Aura MLS observations of stratospheric water
vapor are presented here for the SEAC[4]RS time period of summer 2013. Aura MLS $H_2O$ has 0.4 ppmv precision at
100 hPa for individual profile measurements, with spatial representativeness of 200 km along line-of-sight
(Schwartz et al., 2013). Results shown here use MLS version 4.2 data, but are not significantly different from the
previous version 3.3. MLS observations over CONUS are at ~14:10 local time (ascending orbit) and~1:20 local time
(descending orbit), with successive swaths separated by ~1650 km. Vertical resolution of the water vapor product is
~3 km in the lower stratosphere (Livesey et al., 2016).

Aura MLS shows a seasonal maximum in water vapor over CONUS in July and August. The histogram of Aura
MLS water vapor in Figure 4 indicates that the July-August 2013 CONUS lower stratosphere was drier than the
previous nine-summer MLS record (2004 to 2012). Nevertheless, enhanced lower stratospheric water vapor was
observed by MLS in 2013 as rare but detectable events. From the MLS histogram, the frequency of 100-hPa $H_2O$ >
8ppmv was 0.9% of the observations in July-August 2013 in the blue shaded box. Figure 5 shows that, out of all
MLS 100-hPa water vapor retrievals over the two-month period July to August 2013, water greater than 8 ppmv was
measured only nine times over North America (in the blue shaded box), three times near the west coast of Mexico,
and once over the Caribbean Sea.

**3. Analysis**
Here we briefly describe the analytical technique used to determine whether back trajectories from the aircraft
location intersect OT as identified by a satellite OT data product.

**3.1 Detection of overshooting tops**
In order to link the stratospheric water vapor encountered by the aircraft to the storm systems from which they may
have originated, it is necessary to have a comprehensive continental scale catalog of deep convection. Geostationary
Operational Environmental Satellite (GOES) infrared imagery is used to assemble a catalog of OTs throughout the
U.S. and offshore waters. This catalog was acquired from the NASA LaRC Airborne Science Data From
Atmospheric Composition data archive (http://www-air.larc.nasa.gov/cgi-bin/ArcView/seac4rs). Because OTs are
correlated with storm intensity, the OT product was primarily developed to benefit the aviation community for more
accurate turbulence prediction, as well as the general public for earlier severe storm warnings. However, the product
is also ideally suited for identifying storm systems that can moisten the stratosphere.

Infrared brightness temperatures are used to detect cloud top temperature anomalies within thunderstorm anvils. OT
candidates are colder than the mean surrounding anvil, with the temperature difference indicative of both the
strength of the convective updraft and the depth of penetration. For a description of the method, the reader is
directed to Bedka et al. (2010). The horizontal spatial resolution of the OT product is dependent on the underlying
satellite imagery resolution, i.e., the size of the GOES IR pixel, which is 7 km or less over the CONUS. Additional
validation of OTs requires comparison with the Global Forecast System (GFS) Numerical Weather Prediction
(NWP) model tropopause temperature. The maximum OT cloud height was derived based on knowledge of the 1)
OT-anvil temperature difference, 2) the anvil cloud height based on a match of the anvil mean temperature near to
the OT and the GFS NWP temperature profile, and 3) a temperature lapse rate within the UTLS region based on a
GOES-derived OT-anvil temperature difference and NASA CloudSat OT-anvil height difference for a sample of
direct CloudSat OT overpasses (Griffin et al., 2016). Griffin et al. (2016) finds that 75% of OT height retrievals are
within 0.5 km of CloudSat OT height, so we conservatively estimate the accuracy of the OT altitude to 0.5 km. For
SEAC[4]RS, every available GOES-East and GOES-West scan (typically 15 min resolution) was processed for the
full duration of the mission, even for the non-flight days, yielding a detailed and comprehensive picture of the
location, timing, and depth of penetration of convective storms over the entire CONUS. The output files include the
OT coordinates, time, overshooting intensity in degrees K – which is related to the temperature difference between
the OT and the anvil – and an estimate of maximum cloud height for OT pixels in meters.

The ability of GOES-East and GOES-West to observe an OT depends on its lifetime. OTs are transient events with
lifetimes typically less than 30 minutes but can exceed an hour in well-organized storms such as mesoscale
convective systems and supercell storms (Bedka et al. 2015; Solomon et al., 2016, and references therein).
Animations such as the following show the variability of OTs sampled by GOES at 1-min resolution,
Infrared wavelength animation:
http://cimss.ssec.wisc.edu/goes/srsor2015/800x800_AGOES14_B4_MS_AL_IR_animated_2015222_191500_182_
2015223_131500_182_IR4AVHRR2.mp4
Visible wavelength animation:
http://cimss.ssec.wisc.edu/goes/blog/wp-content/uploads/2015/08/150811_goes14_visible_srsor_MS_mcs_anim.gif
It is clear that some OTs are quite persistent and are both prominent and detectable in IR imagery, but the majority
of OTs in these particular animations are short lived (< 10 minutes). Within these OTs, strong convective updrafts
can transport ice to 16-18 km altitude where turbulent processes such as gravity wave breaking mix tropospheric and
stratospheric air (e.g., Mullendore et al., 2009, 2005; Wang 2003; Homeyer et al. 2017), enabling detrainment of ice
and stratospheric hydration.

Bedka et al. (2010) showed that the OT detection algorithm has a false positive rate of 4.2% to 38.8%, depending
on the size of the overshooting and algorithm settings. As noted above, OTs are transient and can evolve quite
rapidly. The storm top characteristics and evolution we see in the GOES data featured in this paper only capture a
subset of the storm lifetimes, even if we were to have a 100% OT detection rate, due to the 15 min resolution of the
GOES imager. In addition, relatively coarse GOES spatial resolution (up to 7 km over northern latitudes of the US)
can cause the Bedka et al. (2010) method to miss some small diameter and/or weak OT regions. We would be able
to better map storm updraft tracks using data at 1-minute frequency like that shown by Bedka et al. (2015), but this
data is not available over broad geographic domains required for our analysis. Given uncertainties in back
trajectories, GOES under-sampling, and that many OTs can be located in close proximity to one another, we are not
able to make a direct connection between an individual OT and a stratospheric water vapor plume observed a day or
more later. Rather, our analysis identifies a cluster of storms that are the best candidates for generating ice that
sublimates into enhanced water vapor plumes sampled by the ER-2.

**3.2 Back trajectory modeling**
Back trajectories were run from each flight profile where enhanced water vapor was measured to determine whether
the sampled air was convectively influenced. The trajectories were run with the FLEXPART model (Stohl et al.,
2005) using NCEP Climate Forecast System version 2 (CFSv2) meteorology (Saha et al., 2014), and the trajectory
time step interval was one hour. Trajectories were initialized every second along the flight track profiles and run
backward for seven days. A sampled air parcel was determined to be convectively influenced if the back trajectory
from that parcel intercepted an OT region. The tolerances for a trajectory to be considered to have intercepted an OT
cloud were +/-0.25 degrees latitude and longitude, +/-3 hours, +/-0.5 km in altitude. These tolerances were chosen
primarily due to the resolution of the NCEP meteorology used to run the trajectories (1 deg x 1 deg) and also based
on personal communication with Leonard Pfister.

**4. Case Studies**
In this section, we highlight three NASA ER-2 flights where elevated stratospheric water was observed by JLH
Mark2. These dates are 8, 16 and 27 August 2013. Similar results are seen from other hygrometers on the NASA
ER-2 aircraft (J. B. Smith, pers. comm.). For each of these ER-2 flights, the back trajectories are presented along
with the intersection of coincident OT. The cases are described below.

**4.1 First case: 8 August 2013**
Figure 6 shows details of the 8 August 2013 ER-2 aircraft flight. This flight was the transit flight from Palmdale,
California (34.6 °N, 118.1 °W), to Ellington Field, Houston, Texas (29.6 °N, 95.2 °W). In addition to sending the
NASA ER-2 aircraft to the destination base, the science goal of this flight was to profile the North American
Monsoon region with five profiles plus the aircraft ascent and final descent. This flight shows a dramatic transition
from west to east of background stratospheric water to enhanced water. In the lowermost stratosphere (350 K < θ <
380 K), water can be highly variable, but at 90 hPa it is generally unusual to observe water vapor greater than 6
ppmv. As shown in Figure 6c, there is a gradient in water vapor from west to east: 4.0 to 4.4 ppmv at 90 hPa (17
km) over the west coast of CONUS (black and blue points), and greater than 10 ppmv at 90 hPa over Texas (green
points). Simultaneous Aura MLS retrievals also demonstrate a west-to-east water vapor gradient on this day (lines
and filled circles in Figure 6c). Both JLH Mark2 and Aura MLS water vapor exceed the thresholds for enhanced
water vapor.
Analysis of the 8 August 2013 case is shown in Figure 7. For clarity only some example trajectories (a subset of our
analysis) are shown. These are displayed as thin blue traces in panels (b) and (c). The intersections of the example
trajectories with coincident OT are shown as red squares in panels (b) and (c). All overshooting convective tops
within +/-3 hours of the red squares are shown by green symbols in panels (b) and (c). Back trajectories from the
flight track follow the anticyclonic NAM circulation over Western Mexico, Great Plains and Mississippi Valley
(Figure 7b). Every one of the example back trajectories intersects OT, as shown by red symbols in Figure 7b. For
this flight, coincidences with overshooting convection are dominated by overshooting clouds over the Mississippi
Valley and Great Plains. All overshooting convection within the tolerances prescribed (see Sect. 3.2) for the back
trajectories are shown by the green symbols in Figure 7b and 5c. Figure 7c demonstrates the range of altitudes
reached by the coincident overshooting convection and how many convective overshooting cells were coincident.
The high resolution of the convective overshooting data meant that there could be multiple coincident convective
overshooting cells for a single location on a back trajectory. It is significant that some of the green overshooting
cells are higher altitude than the red coincident points, suggesting that overshooting air parcels descended slightly
before mixing with the surrounding air. Figure 7d indicates the source of enhanced water was dominated by
overshooting clouds within seven days prior to intercept by the aircraft.

**4.2 Second case: 16 August 2013**
The NASA ER-2 flight of 16 August 2013 was designed to survey the North American Monsoon in a triangular
flight path from Houston, Texas, to the Imperial Valley in Southern California, to Southeastern Colorado and back
to Texas. The NASA ER-2 aircraft performed six dives, encountering enhanced stratospheric water at 16 to 17 km
altitude (Figure 8a). As shown in Figure 8b, back trajectories intersect overshooting tops over the South Central U.S.
(Texas, Oklahoma, Arkansas) and also over the Sierra Madre Occidental mountain range on the west coast of
Mexico. This case is an example of the classic North American Monsoon circulation with a moisture source over the
Sierra Madre Occidental. Anticyclonic transport carried the moisture north from Mexico, and counter-clockwise
around the high pressure (Figure 8b). The altitude range of the convective overshoot is typically 16 to 17 km
altitude, as shown in Figure 8c. The time between OT and intercept by the aircraft ranges from two to seven days
(Figure 8d).

**4.3 Third case: 27 August 2013**
The 27 August 2013 flight performed six dives to sample the North American Monsoon. Stratospheric water was
enhanced to 15 to 20 ppmv in altitudes ranging from 16.0 to 17.5 km (Figure 9a). The ER-2 aircraft intercepted
highly enhanced stratospheric water from a mesoscale convective complex over the Upper Midwest, which had
overshooting tops over Northern Minnesota and Northern Wisconsin (Toon et al., 2016), as shown in Figure 9b.
Figure 9c shows an abundance of OT above 17 km (green). Generally speaking, the OT appear at higher altitudes in
the northern CONUS/southern Canada than in the Central CONUS. Figure 9d shows that the air masses were
sampled *in situ* by the ER-2 aircraft over Illinois and Indiana one to two days after the intense storm. As is a
common theme for all these experiment days, a portion of the back-trajectories also trace back to overshooting tops
over the Sierra Madre Occidental one week prior.

**5. Conclusions**
In this paper we have examined *in situ* measurements of stratospheric water by JLH Mark2 on the ER-2 aircraft
during the SEAC[4]RS field mission. With JLH Mark2 data, enhanced $H_2O$ above background mixing ratios was
frequently encountered in the overworld stratosphere between 16 and 17.5 km altitude. Back trajectories initialized
initialized at every 1-sec time stamp along the aircraft flight track at 16 to 17.5 km connect the sampled air parcels
to convective OT within seven days prior to the flight. The trajectory modeling indicates that the identified OT are
associated with larger storm systems over the Central U.S. (Figure 7), deep convection over the Sierra Madre
Occidental (Figure 8), and deep convection over the Upper Midwest U.S. and South Central Canada (Figure 9). For
all the back trajectories in this analysis, the fraction that connect to OT within the previous seven days ranges from
30% to 70% (Figure 10).

The concentrations of enhanced water and the connection to OT suggests a mechanism for moistening the CONUS
lower stratosphere: ice is irreversibly injected into the overworld stratosphere by the most intense convective tops.
The temperatures of the CONUS lower stratosphere are sufficiently warm to sublimate the ice, producing water
vapor mixing ratios elevated 10 ppmv or more above background levels. The summertime CONUS has a high
frequency of thunderstorms with sufficient energy to transport ice to the upper troposphere (Koshak et al., 2015, and
references therein). On rare occasion, these storms have sufficient energy to loft ice through the tropopause and into
the stratosphere. Further evidence of ice is provided by water isotopologues. Evaporation and condensation are
fractionating processes for isotopologues, especially HDO relative to $H_2O$ (e.g., Craig, 1961; Dansgaard, 1964).
Condensation preferentially concentrates the heavier HDO isotopologue, so lofted ice is relatively enriched in
$HDO/H_2O$ compared to gas phase (e.g., Webster and Heymsfield, 2003, and references therein). Ice sublimation is
supported by the enriched $HDO/H_2O$ isotopic signature observed by the ACE satellite over summertime North
America (Randel et al., 2010). Cross-tropopause transport is a consequence of turbulent mixing at cloud top,
possibly enhanced by the existence of breaking gravity waves often occurring near overshooting cloud tops. (Wang,
2003). This study addresses a primary goal of the SEAC[4]RS field mission (Toon et al., 2016), answering
affirmatively the science question: "Do deep convective cloud systems locally inject water vapor and other
chemicals into the overworld stratosphere over the CONUS?" This water is almost certainly injected in the ice
phase and subsequently sublimated in the relatively warm stratosphere over CONUS, leading to irreversible
hydration. From this study, we conclude that the depth of injection was typically 16 to 17.5 km altitude for these
particular summertime events.

Satellite retrievals of water vapor from Aura MLS provide a larger-scale context. The fraction of Aura MLS
observations at 100 hPa (approximately 17 km altitude) with $H_2O$ greater than the 8 ppmv threshold is 0.9% for
July-August 2013. In comparison, Schwartz et al. (2013) reports that, for the nine-year record 2004-2012, July and
August had 1.4% and 3.2% of observations exceed 8 ppmv, respectively. This reinforces the conclusion of Randel et
al. (2015) that OT play a minor role in the mid-latitude stratospheric water budget. At the 100-hPa level in the lower
stratosphere, the year 2013 was slightly drier than the average of 2004-2012 summers (Figure 4). Despite the
relatively dry conditions of summer 2013, there was sufficient enhanced water to be clearly observed in the Aura
MLS retrievals (Figures 4, 5, 6). Limb measurements from Aura MLS come from a ~200 km path through the
atmosphere with ~3 km vertical resolution in the lower stratosphere (Livesey et al., 2016). The aircraft profiles of
water vapor are very similar on ascent and descent profiling (Figure 6c), which allows us to estimate the horizontal
length of these features as greater than 180 km, and a vertical thickness of ~0.5 km. This size is sufficiently large
that the MLS retrieval is sensitive to enhanced water, as shown in Figure 6c.

*In situ* measurements probe on a small-scale air parcels that can be connected to OT that inject ice and, to a lesser
extent, trace gases to the stratosphere (e.g., Ray et al., 2004; Hanisco et al., 2007; Jost et al, 2004). In contrast,
modeling studies tend to focus on large-scale processes. Dessler et al. (2002) and Corti et al. (2008) concluded that
OT are a significant source of water vapor in the mid-latitude lower stratosphere. In contrast, Randel et al. (2015)
used Aura MLS observations to conclude that circulation plays a larger role than OT in controlling mid-latitude
stratospheric water vapor in the NAM monsoon region. Our study shows clear evidence of observable perturbations
to stratospheric water vapor on ER-2 aircraft flights that targeted convectively-influenced air during SEAC[4]RS. In
future work, we plan more detailed back trajectory analysis of air parcels over summertime North America to better
understand the transport of ice and water in the lower stratosphere.
**Code Availability**
**n/a**

**Data Availability**
Data discussed in this manuscript are publically available. The NASA aircraft data are available through the
following digital object identifier (DOI): SEAC4RS DOI 10.5067/Aircraft/SEAC4RS/Aerosol-TraceGas-Cloud.

**Appendices: none**

**Supplement link: none**

**Team List:**
Robert L. Herman[1], Eric A. Ray[2], Karen H. Rosenlof[2], Kristopher M. Bedka[3], Michael J.
Schwartz[1], William G. Read[1], Robert F. Troy[1], Keith Chin[1], Lance E. Christensen[1], Dejian Fu[1],
Robert A. Stachnik[1], T. Paul Bui[4], Jonathan M. Dean-Day[5]
[1]Jet Propulsion Laboratory, California Institute of Technology, Pasadena, California, USA.
[2]National Oceanic and Atmospheric Administration (NOAA) Earth System Research Laboratory (ESRL) Chemical
Sciences Division, Boulder, Colorado, USA.
[3]NASA Langley Research Center, Hampton, Virginia, USA.
[4]NASA Ames Research Center, Moffett Field, California, USA.
[5]Bay Area Environmental Research Institute, Sonoma, California, USA.

**Author Contribution**
Robert Herman prepared the manuscript with contributions from all coauthors and was responsible for all aspects of
the JLH Mark2 as principal investigator. Eric Ray and Karen Rosenlof provided trajectory calculations and
interpretation. Kristopher Bedka provided an overshooting top data product and interpretation. Robert Troy, Robert
Stachnik and Keith Chin operated the JLH Mark2 instrument in the field and downloaded data. Robert Stachnik,
Dejian Fu, Lance Christensen and Keith Chin also developed software components for the JLH Mark2 instrument.
Michael Schwartz and William Read provided Aura MLS data and statistical analysis. T. Paul Bui and Jonathan
Dean-Day measured pressure and temperature with the MMS instrument and provided data.

**Competing interests:**
The authors declare that they have no conflict of interest.

**Disclaimer**

**Acknowledgements**
We thank Jose Landeros and Dave Natzic for providing technical support in the laboratory and field, and the aircraft
crew and flight planners for making these measurements possible. The JLH Mark2 team is supported by the NASA
Upper Atmosphere Research Program and Radiation Sciences Program. Part of this research was performed at JPL,
California Institute of Technology, under a contract with NASA.

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

Contributions of Stratospheric Water Vapor to Decadal Changes in the Rate of Global Warming, Science,
327(5970), 1219-1223, doi:10.1126/science.1182488, 2010.
Stohl, A., C. Forster, A. Frank, P. Seibert, and G. Wotawa: Technical note: The Lagrangian particle dispersion
model FLEXPART version 6.2, Atmos. Chem. Phys., 5, 2461-2474, doi: 10.5194/acp-5-2461-2005-
supplement, 2005.
Toon, O. B., H. Maring, J. Dibb, R. Ferrare, D. J. Jacob, E. J. Jensen, Z. J. Luo, G. G. Mace, L. L. Pan, L. Pfister, K.
H. Rosenlof, J. Redemann, J. S. Reid, H. B. Singh, A. M. Thompson, R. Yokelson, P. Minnis, G. Chen, K. W.
Jucks, and A. Pszenny: Planning, implementation and scientific goals of the Studies of Emissions and
Atmospheric Composition, Clouds and Climate Coupling by Regional Surveys (SEAC[4]RS) field mission, *J.*
*Geophys. Res., 121*, 4967-5009, doi:10.1002/2015JD024297, 2016.
Voemel, H., et al.: Balloon-borne observations of water vapor and ozone in the tropical upper troposphere and lower
stratosphere, *J. Geophys. Res.*, 107(D14), pages ACL 8-1-ACL 8-16, DOI:10.1029/2001JD000707, 2002.
Available online at http://onlinelibrary.wiley.com/doi/10.1029/2001JD000707/full#jgrd8979-fig-0017, 2002.
Wang, P. K., Moisture plumes above thunderstorm anvils and their contributions to cross-tropopause transport of
water vapor in midlatitudes, *J. Geophys. Res., 108*(D6), 4194, doi:10.1029/2002JD002581, 2003.
Webster, C. R., and Heymsfield, A. J.: Water Isotope Ratios D/H, $^{18}O/^{16}O$, $^{17}O/^{16}O$ in and out of Clouds Map
Dehydration Pathways, *Science*, 302, 1742, DOI: 10.1126/science.1089496, 2003.
**TABLES**

**Table 1**  Summary of enhanced water vapor measurements in the overworld stratosphere during SEAC[4]RS*. Dates
are NASA ER-2 aircraft flight dates in day-month-year format, and JLH Mark2 maximum water vapor mixing ratios
(ppmv) are shown for potential temperatures greater than 400 K (left) and in the range 380-400 K (right).

| Date | max. water (ppmv) above 400 K | Pot. Temp. (K) above 400 K | Altitude (km) above 400 K | max. water (ppmv) 380-400K | Pot. Temp. (K) 380-400K | Altitude (km) 380-400K |
|---|---|---|---|---|---|---|
| 8-Aug-2013 | 10.1 | 401.2 | 17.29 | 11.2 | 385.7 | 17.10 |
| 12-Aug-2013 | 8.0 | 400.1 | 17.08 | 13.2 | 388.1 | 16.86 |
| 14-Aug-2013 | 7.7 | 402.2 | 17.38 | 10.7 | 387.4 | 16.75 |
| 16-Aug-2013 | 7.0 | 400.2 | 17.14 | 12.2 | 387.3 | 16.82 |
| 27-Aug-2013 | 15.3 | 402.8 | 17.32 | 17.7 | 380.8 | 16.12 |
| 30-Aug-2013 | 9.2 | 400.2 | 17.27 | 12.0 | 390.0 | 16.81 |
| 2-Sep-2013 | 8.0 | 400.3 | 17.07 | 13.0 | 380.3 | 16.28 |
| 4-Sep-2013 | 6.3 | 405.0 | 17.57 | 10.8 | 380.2 | 16.32 |
| 6-Sep-2013 | 6.8 | 400.1 | 17.12 | 15.6 | 381.0 | 16.32 |
| 11-Sep-2013 | 7.7 | 400.2 | 17.13 | 10.2 | 381.0 | 16.22 |
| 13-Sep-2013 | 6.9 | 401.8 | 17.55 | 9.2 | 382.4 | 16.41 |

* SEAC[4]RS = Studies of Emissions and Atmospheric Composition, Clouds and Climate Coupling by Regional
Surveys

**FIGURE CAPTIONS**

**Figure 1.** JLH Mark2 stratospheric water vapor profiles from 23 aircraft flights during SEAC[4]RS. This altitude
range includes the overworld stratosphere (potential temperature greater than 380 K) and lowermost stratosphere
(tropopause to 380 K). The majority of observations have mixing ratios less than 10 ppmv in the lowermost
stratosphere and less than 6 ppmv in the overworld stratosphere. Enhanced water measurements are the extreme
outliers with high water mixing ratios, with a threshold value of mean plus two standard deviations.

**Figure 2.** Distribution of JLH Mark2 water vapor at potential temperatures 380 K to 400 K in the overworld
stratosphere for all flights in the SEAC[4]RS mission (summer 2013). These potential temperatures correspond to
approximately 16.8 to 17.4 km altitude (99 to 90 hPa).

**Figure 3.** Distribution of JLH Mark2 water vapor at potential temperatures 400 K to 420 K in the overworld
stratosphere for all flights in the SEAC[4]RS mission (summer 2013). These potential temperatures correspond to
approximately 17.4 to 18.0 km altitude (90 to 80 hPa).

**Figure 4.** Distribution of Aura MLS v4.2 100-hPa $H_2O$ over CONUS (blue shaded box in insert), corresponding to
approximately 17 km altitude. The two histograms for July-August 2013 (blue asterisks and trace) and the previous
nine-summer MLS record, July-August 2004 through 2012 (red circles and trace) indicates that 2013 was drier than
average. The threshold for MLS-detected 'enhanced water vapor' (thick black vertical line) is set at 8 ppmv, same as
Schwartz et al. (2013), to exclude the larger population of measurements at 6 to 8 ppmv water vapor that may have
other sources.

**Figure 5.** Two-month mean map of Aura MLS v4.2 100-hPa $H_2O$ (color scale), corresponding to approximately 17
km altitude, with superimposed MERRA horizontal winds (arrows) for July-August 2013 during the SEAC[4]RS time
period. MLS observations of 100-hPa $H_2O$ greater than 8 ppmv in this two-month period are shown by the white
circles.

**Figure 6.** Map and profiles of aircraft and satellite water vapor on 8 August 2013 over California (number 1 shown
in dark blue) and Texas (number 2 shown in green). (a) Map of ER-2 aircraft flight track (solid colored trace) and
nearly coincident Aura MLS geolocations (asterisks and lines). (b) ER-2 aircraft altitude profiles (solid colored
trace) color-coded by dives and MLS times (horizontal lines). (c) Vertical profiles of *in situ* water vapor
measurements from JLH Mark2 (dots) and MLS retrievals of water vapor (circles and lines). Some measurements
exceed the threshold for enhanced water vapor of 8 ppmv for Aura MLS (after Schwartz et al., 2013), and the
campaign-wide mean plus 2 st. dev. for JLH Mark 2, 9.7 ppmv at 380-400 K and 6.6 ppmv at 400-420 K.

**Figure 7.** Analysis of the 8 August 2013 NASA ER-2 aircraft flight. (a) Vertical profiles of JLH Mark2 *in situ* $H_2O$.
Back trajectories were initialized from all aircraft water measurements at 16 to 17.5 km altitude. (b) Example back
trajectories (thin blue traces) and coincident overshooting convection (red). Along the NASA ER-2 flight track
(orange line), enhanced water vapor was measured (thick blue lines). This figure identifies where trajectories and
OT are coincident (red squares) within tolerances prescribed in Section 3.2. The green markers are overshooting
convective tops within +/-3 hours of the red squares to indicate the main regions of convective overshooting during
the seven days prior to the ER-2 flight and which of those regions appeared to contribute most to the water vapor
enhancement measured on the flight. (c) Altitude plot of example back trajectories showing coincident overshooting
(red squares). The green markers are overshooting convective tops within +/-3 hours of the red squares. The high
resolution of the convective overshooting data meant that there could be multiple coincident convective
overshooting cells for a single location on a back trajectory, (d) Days between OT and intercept by aircraft on 8
August 2013.

**Figure 8.** Analysis of the 16 August 2013 NASA ER-2 flight. (a) Vertical profiles of JLH Mark2 *in situ* $H_2O$ similar
to Figure 7a, (b) Back trajectories from the aircraft path similar to Figure 7b, (c) Altitude plot of back trajectories
showing coincident overshooting (red) and all overshooting within +/- 3 hours (green) similar to Figure 7c, (d) Days
between OT and intercept by aircraft similar to Figure 7d.

**Figure 9.** Analysis of the 27 August 2013 NASA ER-2 flight. (a) Vertical profiles of JLH Mark2 *in situ* $H_2O$ similar
to Figure 7a, (b) Back trajectories from the aircraft path similar to Figure 7b, (c) Altitude plot of back trajectories
showing coincident overshooting (red) and all overshooting within +/- 3 hours (green) similar to Figure 7c, (d) Days
between OT and intercept by aircraft similar to Figure 7d.

**Figure 10.** Fraction of back trajectories that intersected OTs during the 7 previous days for the three SEAC$^4$RS
flights of 8 August (blue), 16 August (green) and 27 August 2013 (red) shown in Figures 7, 8 and 9, respectively.

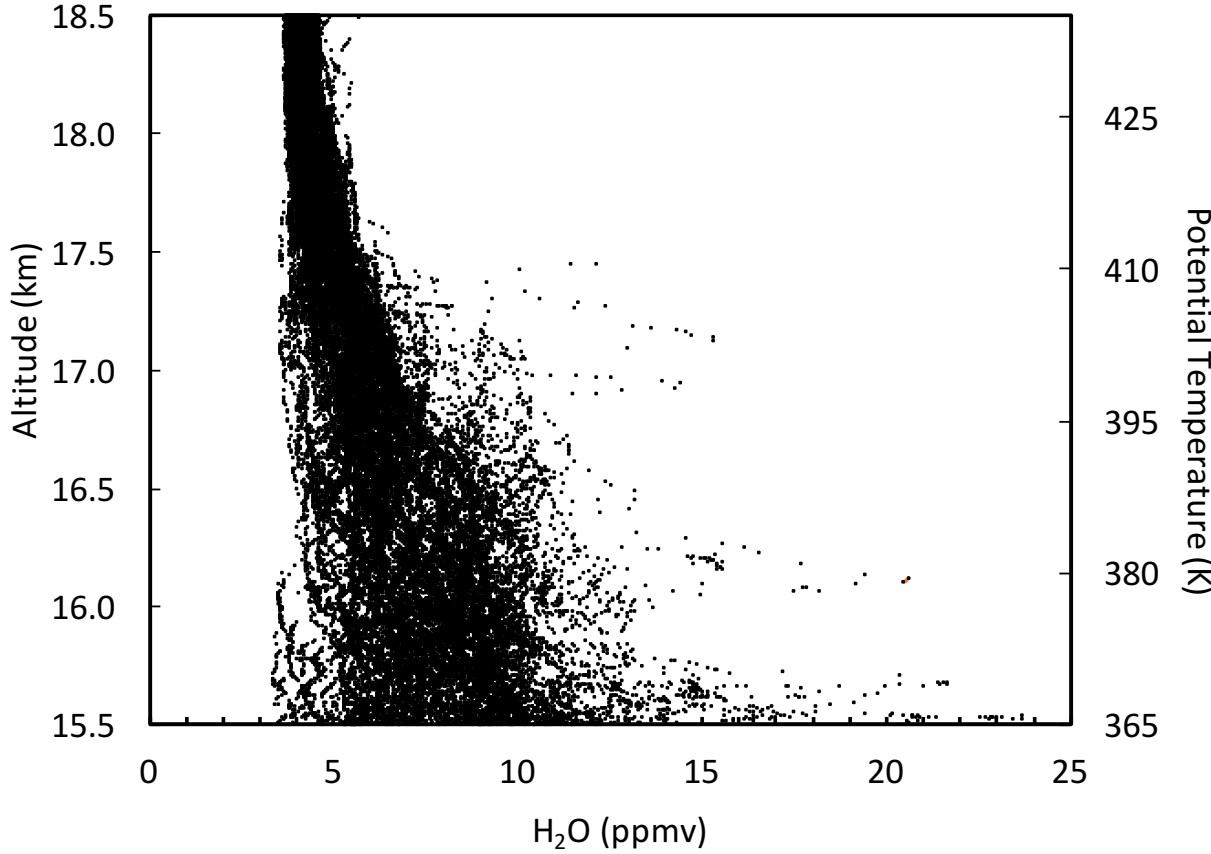


**Figure 1.** JLH Mark2 stratospheric water vapor profiles from 23 aircraft flights during SEAC[4]RS. This altitude
range includes the overworld stratosphere (potential temperature greater than 380 K) and lowermost stratosphere
(tropopause to 380 K). The majority of observations have mixing ratios less than 10 ppmv in the lowermost
stratosphere and less than 6 ppmv in the overworld stratosphere. Enhanced water measurements are the extreme
outliers with high water mixing ratios, with a threshold value of mean plus two standard deviations.

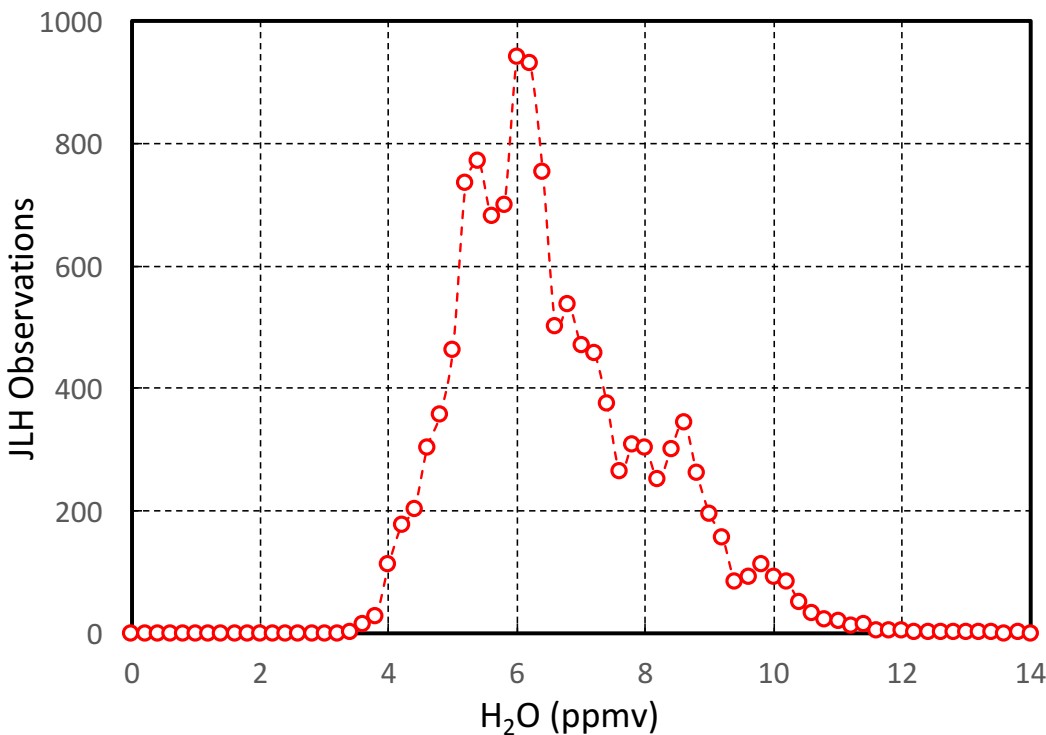


**Figure 2.** Distribution of JLH Mark2 water vapor at potential temperatures 380 K to 400 K in the overworld
stratosphere for all flights in the SEAC[4]RS mission (summer 2013). These potential temperatures correspond to
approximately 16.8 to 17.4 km altitude (99 to 90 hPa).

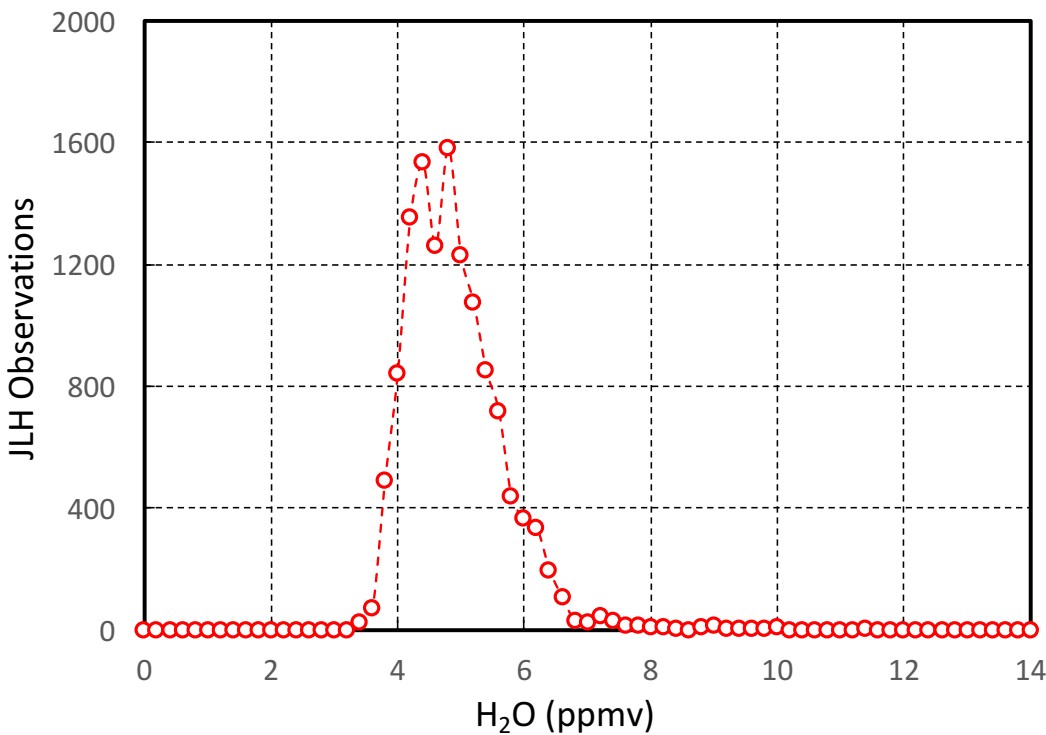

**Figure 3.** Distribution of JLH Mark2 water vapor at potential temperatures 400 K to 420 K in the overworld
stratosphere for all flights in the SEAC[4]RS mission (summer 2013). These potential temperatures correspond to
approximately 17.4 to 18.0 km altitude (90 to 80 hPa).

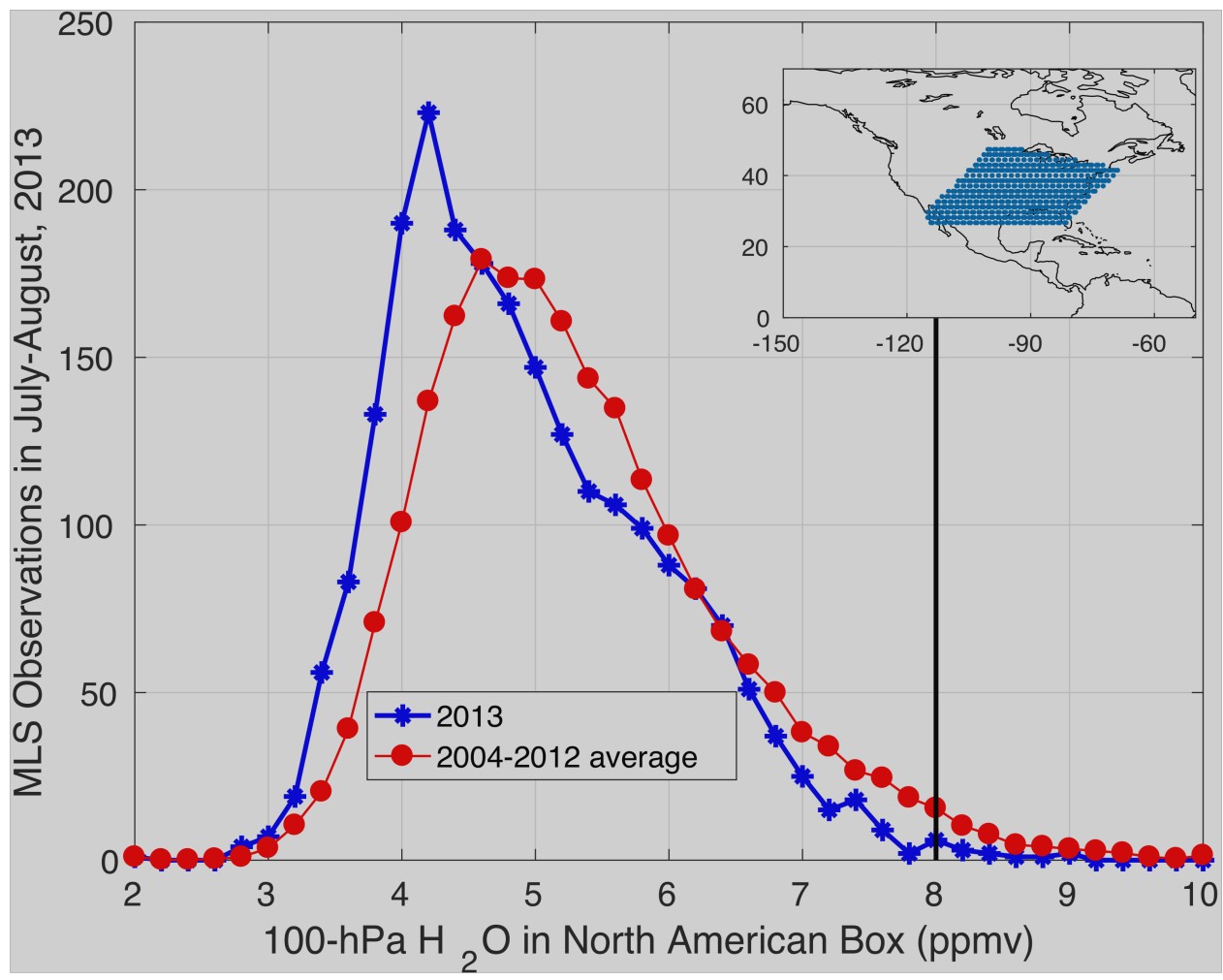

**Figure 4.** Distribution of Aura MLS v4.2 100-hPa $H_2O$ over CONUS (blue shaded box in insert), corresponding to
approximately 17 km altitude. The two histograms for July-August 2013 (blue asterisks and trace) and the previous
nine-summer MLS record, July-August 2004 through 2012 (red circles and trace) indicates that 2013 was drier than
average. The threshold for MLS-detected 'enhanced water vapor' (thick black vertical line) is set at 8 ppmv, same as
Schwartz et al. (2013), to exclude the larger population of measurements at 6 to 8 ppmv water vapor that may have
other sources.

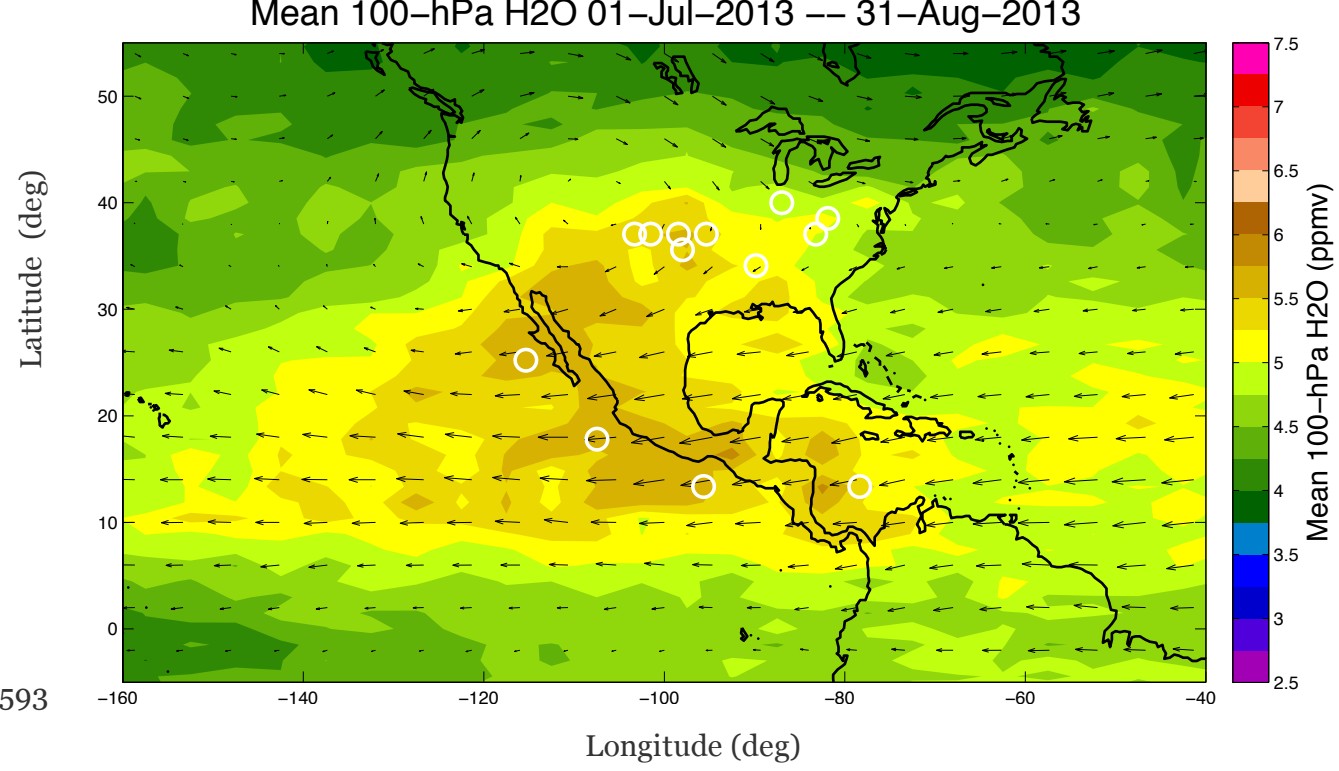

**Figure 5.** Two-month mean map of Aura MLS v4.2 100-hPa $H_2O$ (color scale), corresponding to approximately 17 km altitude, with superimposed MERRA horizontal winds (arrows) for July-August 2013 during the SEAC[4]RS time period. MLS observations of 100-hPa $H_2O$ greater than 8 ppmv in this two-month period are shown by the white circles.

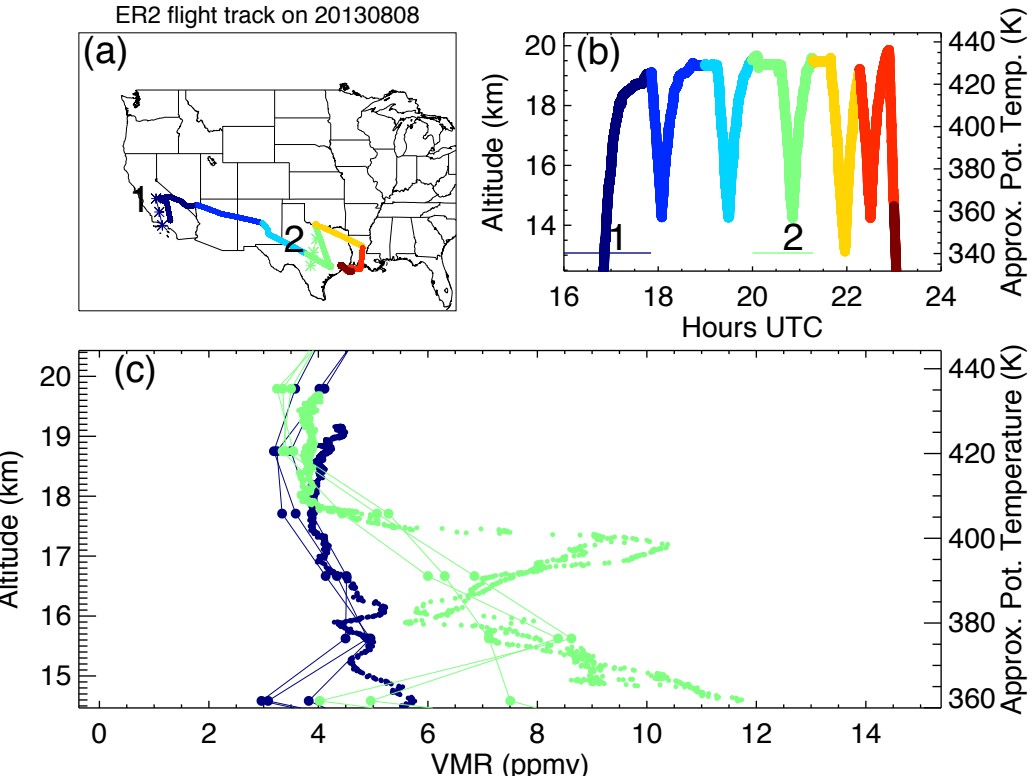

**Figure 6.** Map and profiles of aircraft and satellite water vapor on 8 August 2013 over California (number 1 shown
in dark blue) and Texas (number 2 shown in green). (a) Map of ER-2 aircraft flight track (solid colored trace) and
nearly coincident Aura MLS geolocations (asterisks and lines). (b) ER-2 aircraft altitude profiles (solid colored
trace) color-coded by dives and MLS times (horizontal lines). (c) Vertical profiles of *in situ* water vapor
measurements from JLH Mark2 (dots) and MLS retrievals of water vapor (circles and lines). Some measurements
exceed the threshold for enhanced water vapor of 8 ppmv for Aura MLS (after Schwartz et al., 2013), and the
campaign-wide mean plus 2 st. dev. for JLH Mark 2, 9.7 ppmv at 380-400 K and 6.6 ppmv at 400-420 K.

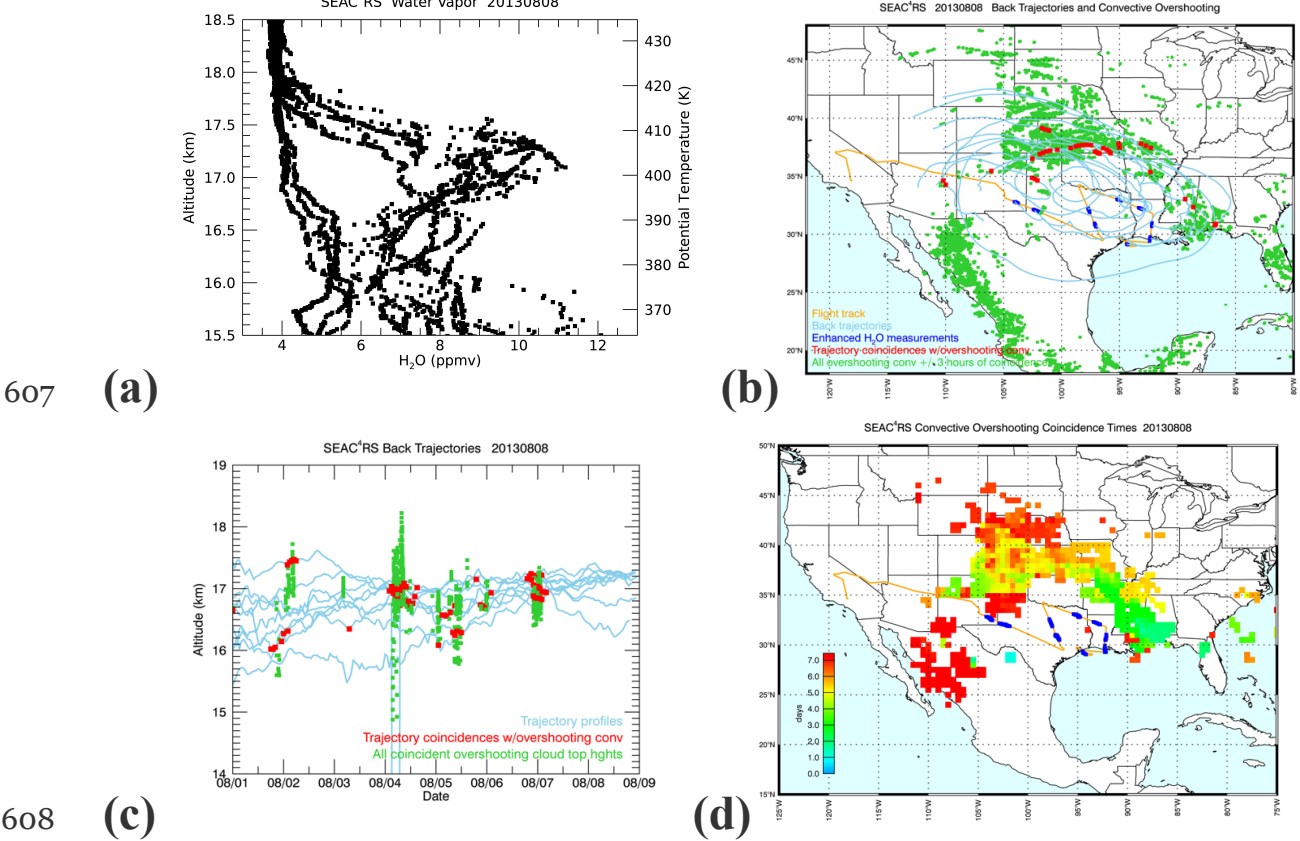

**(a)**                **(b)**
**(c)**                **(d)**
**Figure 7.** Analysis of the 8 August 2013 NASA ER-2 aircraft flight. (a) Vertical profiles of JLH Mark2 *in situ* $H_2O$.
Back trajectories were initialized from all aircraft water measurements at 16 to 17.5 km altitude. (b) Example back
trajectories (thin blue traces) and coincident overshooting convection (red). Along the NASA ER-2 flight track
(orange line), enhanced water vapor was measured (thick blue lines). This figure identifies where trajectories and
OT are coincident (red squares) within tolerances prescribed in Section 3.2. The green markers are overshooting
convective tops within +/-3 hours of the red squares to indicate the main regions of convective overshooting during
the seven days prior to the ER-2 flight and which of those regions appeared to contribute most to the water vapor
enhancement measured on the flight. (c) Altitude plot of example back trajectories showing coincident overshooting
(red squares). The green markers are overshooting convective tops within +/-3 hours of the red squares. The high
resolution of the convective overshooting data meant that there could be multiple coincident convective
overshooting cells for a single location on a back trajectory, (d) Days between OT and intercept by aircraft on 8
August 2013.

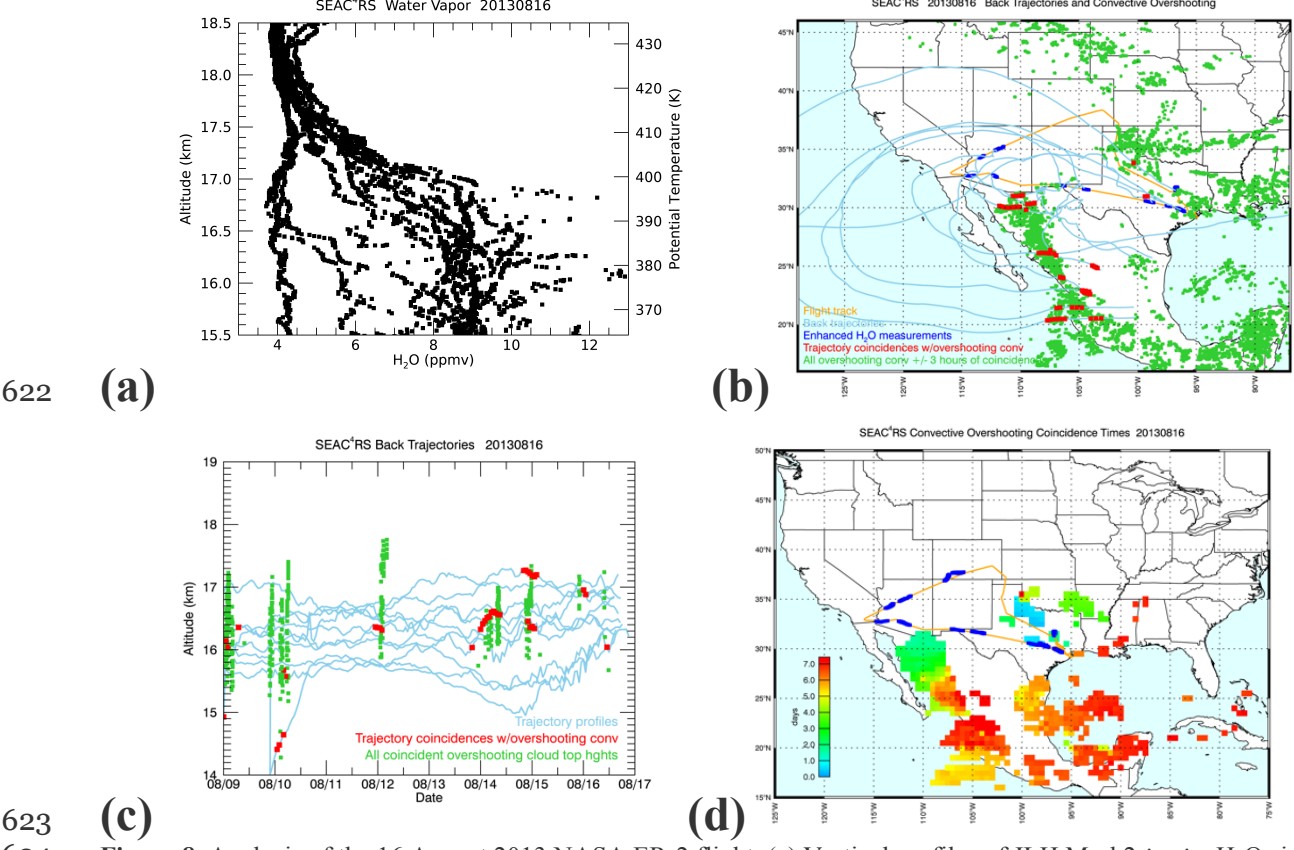

**(a)**                                    **(b)**
**(c)**                                    **(d)**
**Figure 8.** Analysis of the 16 August 2013 NASA ER-2 flight. (a) Vertical profiles of JLH Mark2 *in situ* $H_2O$ similar
to Figure 7a, (b) Back trajectories from the aircraft path similar to Figure 7b, (c) Altitude plot of back trajectories
showing coincident overshooting (red) and all overshooting within +/- 3 hours (green) similar to Figure 7c, (d) Days
between OT and intercept by aircraft similar to Figure 7d.


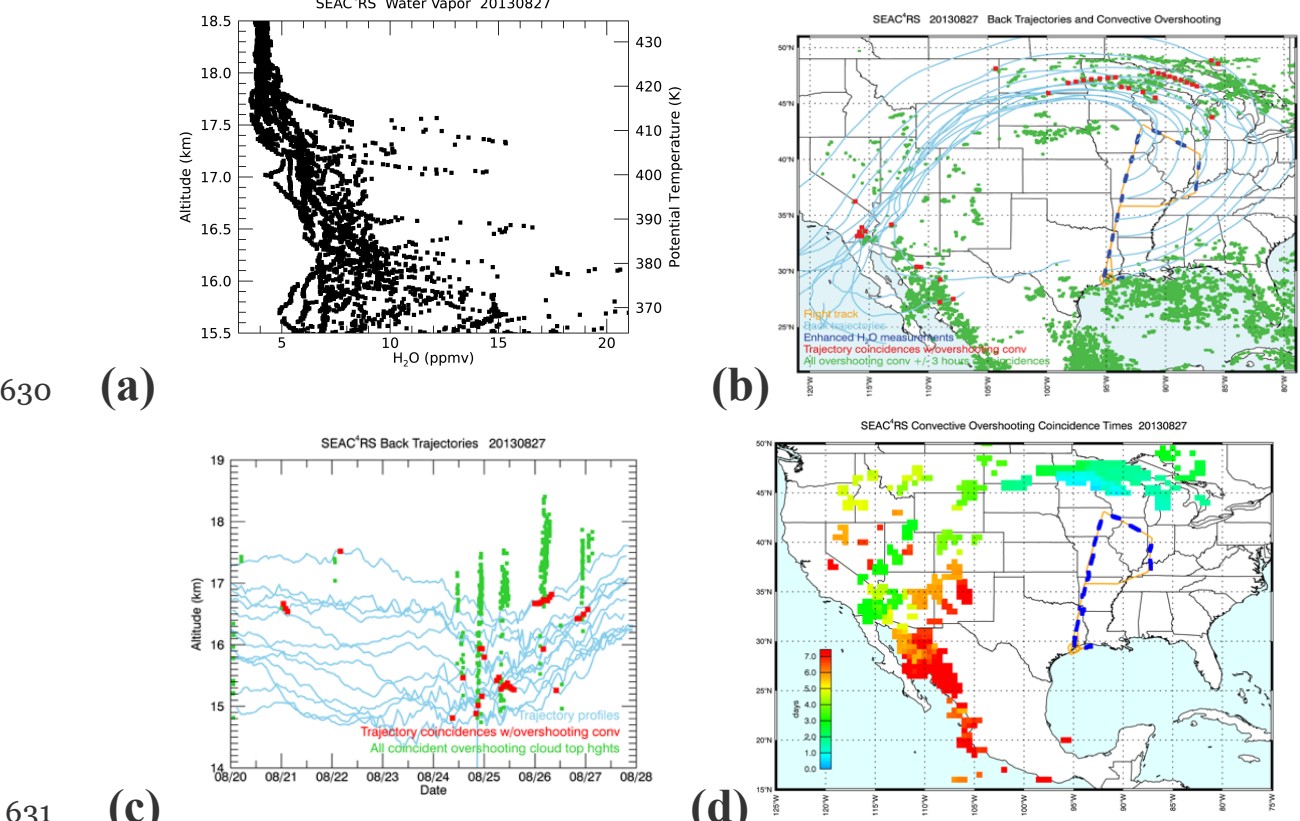

**(a)**
**(c)**
**(b)**
**(d)**

**Figure 9.** Analysis of the 27 August 2013 NASA ER-2 flight. (a) Vertical profiles of JLH Mark2 *in situ* $H_2O$ similar
to Figure 7a, (b) Back trajectories from the aircraft path similar to Figure 7b, (c) Altitude plot of back trajectories
showing coincident overshooting (red) and and all overshooting within +/- 3 hours (green) similar to Figure 7c, (d)
Days between OT and intercept by aircraft similar to Figure 7d.

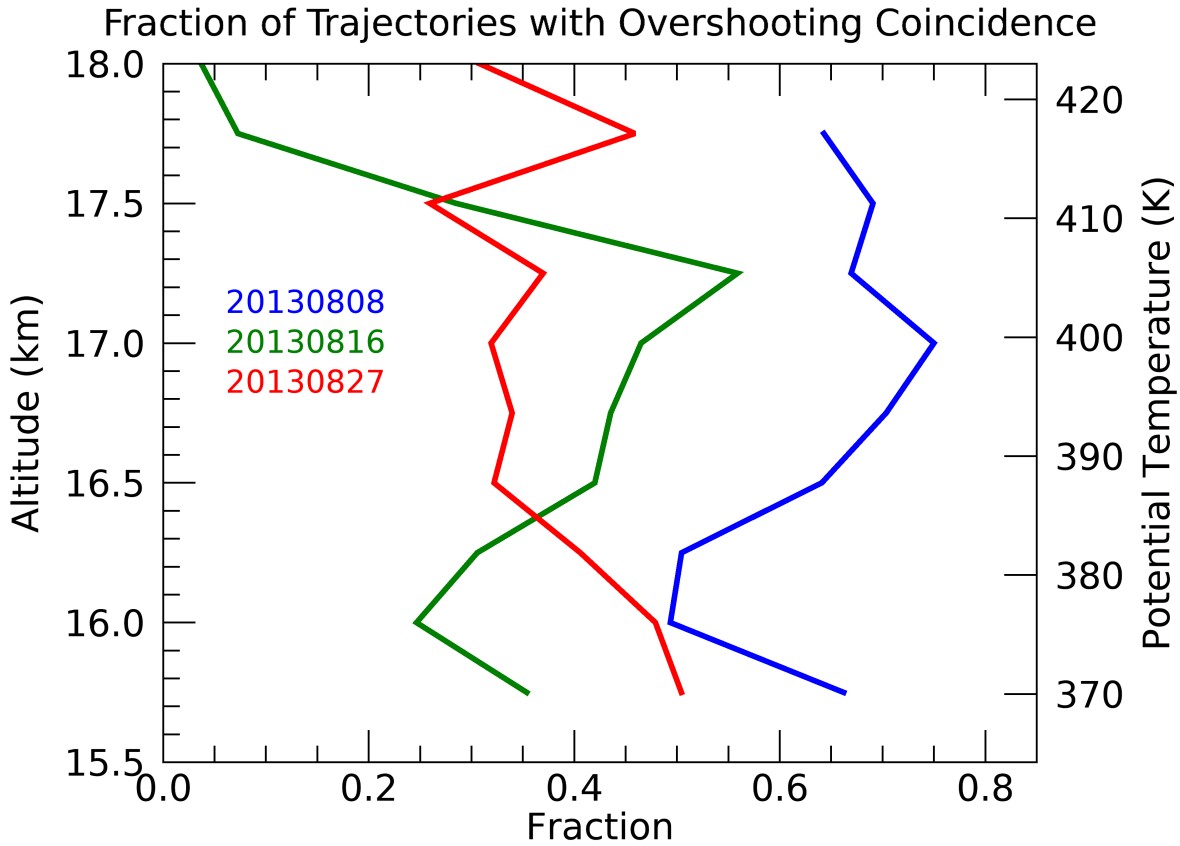

**Figure 10.** Fraction of back trajectories that intersected OTs during the 7 previous days for the three SEAC[4]RS
flights of 8 August (blue), 16 August (green) and 27 August 2013 (red) shown in Figures 7, 8 and 9, respectively.