# Peer review of "Continental United States and the Role of Overshooting Convection"

_Atmospheric Chemistry and Physics, 2016_

## Referee Comment (RC1) · Anonymous Referee #1 · 10 Jan 2017

This paper presents an interesting analysis that links anomalously high stratospheric water vapor mixing ratios measured over the central United States to convection-induced overshooting cloud tops via model-calculated back trajectories of air parcels. High levels of stratospheric water vapor were measured during three flights of the NASA ER-2 aircraft as part of the SEAC4RS mission in August 2013. Trajectories from the FLEXPART model, initiated at the times and locations of the high water vapor measurements and stepped back in time, show convincing spatiotemporal intersections with overshooting cloud tops identified from GOES infrared imagery of cloud brightness temperatures.

This paper is fairly well written but could use some "cleaning up" of the language and

presentation. The methods and conclusions are sensible and the content is appropriate for Atmospheric Chemistry and Physics. I suggest it be published after some minor but important revisions.

Major Comment:

In my opinion this paper could easily stand alone if all Figures and in-depth discussions of Aura MLS stratospheric water vapor measurements were omitted. The vast majority of scientific conclusions can be made without the broader picture provided by the MLS data. The one significant contribution of MLS data to this paper is to show the low frequency of occurrence of anomalously high stratospheric water vapor mixing ratios over the North American monsoon region. However, this conclusion, based on MLS water vapor data, was already published by Schwartz et al. (2013, Geophy. Res. Lett.). There are also some difficulties translating between the aircraft and MLS results because of the presumably different mixing ratio thresholds used to identify "enhanced" water vapor for each. Intuitively the MLS threshold needs to be lower because of the much coarser vertical and horizontal resolution of MLS retrievals. See my comments below regarding this issue.

I will leave it up to the authors if they want to retain or omit the MLS data in their paper. I don't think its presence detracts from the main objectives of this paper, but I also think it doesn't contribute much to them.

Minor Comments:

Figure 1 and Caption. "Each monthly histogram is normalized to unity over mixing ratio" needs further explanation. I went back to the Schwartz et al. (2013) paper for clarification and found the exact same statement. For me, to normalize each monthly histogram one would divide the population of each mixing ratio bin by the entire population or the population of the bin containing the mode or mean of the distribution. Assuming a somewhat Gaussian distribution, the normalization the mode or mean bin population would produce numbers near one and zero for the most and least populated

bins (most and least probable in a PDF), respectively. This is how I interpret Figure 1 even though I don't understand "normalized to unity over mixing ratio" and the units on the color bar are ppmvˆ-1. What am I missing here?

Also for Figure 1, the vertical lines are not "dashed" and the gray shading is too light, at least for my eyes.

The paper uses three vertical coordinates and doesn't tie them together very well. The introduction focuses on potential temperatures, everything pertaining to Figures 1-4 is discussed and shown in pressure coordinates, then Figures 5-7 are presented in altitude coordinates. Are the profiles in Figures 4-7 entirely in the lowermost stratosphere or do they extend into the overworld (or upper troposphere for that matter)? Profiles in Figures 4 extend from 150 to 50 hPa but in Figure 5a (same profiles) span 15.5 to 18.5 km. Are the axis ranges of these two Figures the same in terms of vertical span? I understand the need to discuss stratospheric layers in terms of potential temperature (introduction), but why can't everything else be presented uniformly using either pressure or altitude coordinates (or both together)? It would make the discussion and Figures much more intelligible.

Though the phrase "enhanced water vapor" appears in the paper's title and is used frequently throughout the paper, it is never defined for the aircraft measurements. Presumably there is a mixing ratio threshold used to identify air parcels with enhanced water vapor? The abstract may indirectly imply, likely incorrectly, that "enhanced" mixing ratios measured from aircraft are those >10 ppmv. Adding to the mystery are the blue markers in panels (a) of Figures 5-7 that represent the "enhanced H2O region" but range well below 5 ppmv. What exactly is the threshold for "enhanced" water vapor measured from the aircraft? Why are there mixing ratios <5 ppmv in regions of enhanced water vapor? This is quite confusing and requires some important clarification in the paper.

Is it appropriate to try to combine information from MLS and aircraft-based detections of

"enhanced" water vapor in this paper when their thresholds are probably very different? Figure 3 shows only 13 instances where MLS retrievals at 100 hPa during Jul-Aug 2013 exceeded 8 ppmv. Is this the threshold for MLS-detected "enhanced" water vapor? Given the greatly different vertical and horizontal resolution of the MLS and aircraft measurements, how can the MLS findings be integrated with the aircraft-based results that follow? I don't think they can. What does a MLS retrieval of 8 ppmv with a 3-km averaging kernel width translate to for the in situ aircraft measurements? Figures 1-3 contribute to only one conclusion drawn in this paper: that MLS retrievals at 100 hPa over the NAM region during Jul-Aug rarely exceed 8 ppmv. In my opinion that is a secondary (and already published) conclusion compared to the dominant conclusion of this paper that enhanced stratospheric water vapor measured over the NAM region during Jul-Aug 2013 can be traced back to convection-induced overshooting cloud tops.

Editorial Comments:

Line 58: please give the lowermost stratosphere a rough lower limit (in potential temperature) otherwise this implies it extends down to the surface. "Commonly" suggests these mechanisms are highly probably pathways for tropospheric water vapor to reach the stratosphere, which they are not.

Line 62: Please make it clear that ice (not elevated water) is transported into the stratosphere where it sublimates and produces "enhanced" water vapor mixing ratios.

Line 63: Here and elsewhere "stratospheric overworld" (defined in Line 45) has now become the "overworld stratosphere".

Line 66: Ice does not "bypass" the cold trap, it is unaffected by it.

Line 67: Suggested paragraph break before "Paraphrasing".

Line 81: "are limited by their horizontal and vertical resolution in detecting fine-scale three-dimensional variations in water vapor ..."

Line 101: "Instruments on the NASA ER-2" – which instruments and what was measured that allows you to conclude that "the aircraft intercepted convectively-influenced air"?

Line 107: "measures daily global atmospheric profiles" sounds like one gigantic profile is measured each day. How about "measures ∼3500 profiles around the globe each day of atmospheric species ..."

Line 118: Aren't "the decadal histogram" and "the previous multi-year MLS record" the same thing in Figure 2? Don't you want to compare and contrast the histogram of 2013 with the histogram of 10-year mean values? Figure 2 would be a great place to visually show (as a vertical line) the threshold for "enhanced" MLS water vapor.

Line 120: You have the histograms so why say "rare" when you can be quantitative?

Line 122: At least 3 of the white circles in Figure 3 are near the west coast of Mexico, not Central America.

Line 146: What is the spatial resolution (horizontal and vertical) of the convective storm information used to link "enhanced" water vapor to overshooting tops? Later (Line 503) you say that the OT data are "high resolution" but the spatial resolution is never described.

Line 156: What was the time step interval for back trajectory calculations?

Line 158: How were these tolerances chosen? Do they have any relationship to the spatial resolution (horizontal and vertical) of the convective storm information?

Line 164: You already wrote about initializations (Line 153). This statement is again repeated in Line 180.

Line 169: It would be beneficial to include the lats/lons of these sites.

Line 173: Same comment as for Line 58.

Line 182: This information is already in the Figure Captions.

Line 184: In Figures 5-7, panels (b) and (c), the green markers show "near coincidences" (within the tolerances listed in Line 158) between back trajectories and OTs, while red squares show "coincidences". What are the criteria for "coincidences"?

Line 191: In Figure 5b there are many green markers in western Mexico, so why is this location not listed?

Line 198: In Figure 5d there are many red markers depicting transit times >6 days, but here you claim domination by transit times of "two to five days".

Line 205: TX, OK and AR are in the "South Central" U.S.

Line 216: Define MCC as mesoscale convective complex

Line 227: Earlier you stated that back trajectories were initialized for every measurement of enhanced water vapor. Now, "Example" infers that this was done only for a subset. Did "all" (Line 228) of the back trajectories connect to OTs? Really? All?

Line 230: Figure 7b shows there were influences from storm systems in South Central Canada as well.

Line 235: "leaving behind" should be "producing"

Line 238: "propel water" gives the wrong impression while "loft ice" is more accurate.

Line 239: "the enriched delta-D isotopic signature" needs a bit more explanation, including a mention of the isotope itself, HDO.

Line 246: "The water is almost certainly injected in the ice phase" needs support. I suggest you combined this paragraph with all or some of the previous paragraph that provides such support.

Line 252: "from a long (200 km) path through the atmosphere" also needs to include information about the vertical resolution. The qualitative statement "may be enhanced even more" is the basis for my argument that the MLS and aircraft results cannot be

meaningfully integrated together in this paper.

Line 254: what percent of the 10 years of MLS observations show enhanced water?

Line 261: Which "monsoon region"?

Figure 2: Caption: "ten year mean for 2004 through 2013."

Figure 3: Caption: "Average Aura MLS 100-hPa"

Figure 4: Colored markers are not "retrievals" from the aircraft, they are the actual aircraft measurements. Why do multiple aircraft profiles produce only one profile convolved with the MLS averaging kernels? Without strongly magnifying this Figure I can't tell the difference between the MLS profiles and the convolved aircraft profile. I suggest omitting the convolved aircraft profile in each panel. It does shows that the averaging kernels smooth the aircraft profiles, but isn't that exactly what one would expect anyway? Also, the black asterisks showing MLS retrieval locations on flight track maps are difficult to distinguish from black map lines. Perhaps larger gray symbols "x" or "+" would stand out more? And the "line" mentioned in the caption, is this the horizontal line in each "Flight Altitude" (should be "Pressure") vs UTC hour panel indicating the time range of measurements shown in the profiles? This Figure would be an ideal place to visually show (as a vertical line in each profile panel) the threshold for aircraft "enhanced" water vapor.

Figure 5: These comments apply to each of Figures 5-7: In panels (a), the captions claim the blue markers denote "enhanced water measurements" (which range below 5 ppmv) while in the panel (a) legends the blue markers are said to represent the "Enhanced H2O region", which must be something quite different from "enhanced water vapor" measurements. This distinction needs to be clarified in the paper by defining exactly what is meant by the terms "enhanced water measurements" and "enhanced H2O region".

---

## Referee Comment (RC2) · Anonymous Referee #2 · 15 Jan 2017

Review of "Enhanced Stratospheric Water Vapor over the Summertime Continental United States and the Role of Overshooting Convection" by Herman et al.

This paper presents direct airborne measurements of water injection into the lowermost stratosphere over the continental United States by convective overshooting tops and relates these to individual overshooting events through trajectory analysis.

The study is generally well written, however, the overall result and conclusion is somewhat weak. I would recommend this paper for publication only after major revisions, for which I give suggestions below.

Major comments:

[Figure]

The observations themselves are not new and a number of previous studies have clearly indicated that overshooting convection may transport water ice into the stratosphere, where it evaporates and increases the stratospheric water vapor concentration. The novelty of this study is that it links observed water vapor enhancements to possible overshooting top events through trajectory analysis. This result, while new, is not very surprising and leaves the paper with a rather insignificant result. The paper would benefit strongly from a discussion of the significance of this result and a much enhanced statistical analysis using their entire observational set. The authors indicated that they have many more observations during this campaign but chose to show only three examples. The authors might want to use their entire data set and increase their statistical analysis. Their only statistical argument is at the end of the discussion, where they use only MLS data to state, that the impact is small. However, their own data (Figure 4) shows nicely, that MLS misses the highest concentrations due to its strong vertical averaging, which will heavily skew the result. Since the water vapor enhancements seem present on a very large scale, it would be good to see the entire data set for this campaign. The authors could then attempt to make a statistical analysis on how well they can relate these enhancements to OT events, what their temporal distribution may have been, and if there could be some preferred regions. In the past water vapor instruments onboard the high altitude aircraft have shown significant disagreements. The authors state, that the other instruments show similar results. It would be good to actually show these, which would support the confidence in the observations themselves.

Minor comments:

The manuscript should try to stick to one vertical coordinate and add other vertical coordinates only as additional information, e.g '90 hPa (370 K)'. Figure 4 uses pressure as vertical coordinate for consistency with MLS. Therefore, this could be the vertical coordinate system of choice. The profile figures may add approximate potential temperature as additional vertical axis for reference.

Most data shown in Figure 4 repeat between panels a-c. This figure could be combined

into one panel with MLS data color coded roughly following the aircraft data.

The use of green dots in Figures 5-7 is confusing. Panels c seem to indicate coincidences with relaxed conditions, whereas panels b seem to indicate all overshooting top events in the given time frame to show convective regions. This should be clarified.

There are several references to "stratospheric background levels". How where these background levels defined for this purpose? Are the profiles west of the Rocky Mountains considered "background" or did the authors use something else to define what the background is for this purpose? If they used the West Coast profiles, then they should briefly discuss the meteorology and exclude that these are more typical high latitude profiles. Could it be that the "background" is not as low as the authors assume?

There is obviously a large uncertainty in the detection and assignment of OT events. It would be good if the authors discussed how this uncertainty impacts their identification of possibly source events. What is the lifetime of a typical overshooting top? How many are likely to be missed by the OT detection algorithm? Especially on the events that are closer to the observations, can the authors identify individual events that are best candidates?

Line 118-119: better: ' . . . was drier than the 10 year MLS record . . .'

Line 129-130: better '. . . the storm systems from which they may have originated, it is necessary . . .'
* * *

---

## Author Comment (AC1) · 28 Jan 2017

Authors' response to anonymous referee #1 on "Enhanced Stratospheric Water Vapor over the Summertime Continental United States and the Role of Overshooting Convection" by R. L. Herman et al., ACP-2016-1065

We would like to thank the referee #1 for detailed review, and for insightful and constructive comments (shown in quotations below). The individual points are addressed by the authors below:

Referee's Major Comment: "In my opinion this paper could easily stand alone if all Figures and in-depth discussions of Aura MLS stratospheric water vapor measurements

were omitted. The vast majority of scientific conclusions can be made without the broader picture provided by the MLS data. The one significant contribution of MLS data to this paper is to show the low frequency of occurrence of anomalously high stratospheric water vapor mixing ratios over the North American monsoon region. However, this conclusion, based on MLS water vapor data, was already published by Schwartz et al. (2013, Geophy. Res. Lett.). There are also some difficulties translating between the aircraft and MLS results because of the presumably different mixing ratio thresholds used to identify "enhanced" water vapor for each. Intuitively the MLS threshold needs to be lower because of the much coarser vertical and horizontal resolution of MLS retrievals. See my comments below regarding this issue." "I will leave it up to the authors if they want to retain or omit the MLS data in their paper. I don't think its presence detracts from the main objectives of this paper, but I also think it doesn't contribute much to them."

Authors' Response: The Aura MLS stratospheric water vapor measurements place the aircraft field mission data into regional and decadal perspective. We have improved the text and figures (as described below) to tie MLS and the aircraft results together. MLS helps us address the question from Toon et al (2016): "Do deep convective cloud systems locally inject water vapor and other chemicals into the overworld stratosphere over the CONUS?"

We modified text in lines 116-117 (lines 135-136 in revised manuscript) for greater clarity: "Figure 1 adds the year 2013 including the SEAC4RS time period to the long-term Aura MLS 100-hPa time series from Schwartz et al. (2013)."

Also, in our revised paper we will demonstrate that both MLS water at 100 hPa and aircraft water are enhanced significantly above background. This point will be demonstrated in a clearer Figure 4 and histograms of both MLS and aircraft data.

Referee's Minor Comments: 1) Figure 1 and Caption. "'Each monthly histogram is normalized to unity over mixing Ratio' needs further explanation. I went back to the

[Figure]

Schwartz et al. (2013) paper for clarification and found the exact same statement. For me, to normalize each monthly histogram one would divide the population of each mixing ratio bin by the entire population or the population of the bin containing the mode or mean of the distribution. Assuming a somewhat Gaussian distribution, the normalization the mode or mean bin population would produce numbers near one and zero for the most and least populated bins (most and least probable in a PDF), respectively. This is how I interpret Figure 1 even though I don't understand "normalized to unity over mixing ratio" and the units on the color bar are ppmvËĘ-1. What am I missing here?"

Author's response to 1: We agree with the referee that this was not explained well in the Figure 1 caption. If the integral from 5 ppmv to 8 ppmv in a given month was 0.6, that would mean that 60% of observations fell in that range of mixing ratios. We have modified the Figure 1 caption: "Each monthly histogram is normalized over mixing ratio such that, if one integrates colors (ppmv-1) over a range of mixing ratio, the result is a dimensionless probability that is normalized to one when integrating over all colors for that month."

Figure 1 comment: "Also for Figure 1, the vertical lines are not "dashed" and the gray shading is too light, at least for my eyes."

Author's response to Figure 1: In the Figure 1 caption, we remove the word "dashed" and the gray shading is now darker (see accompanying Figure 1).

Figure 1. Time series of Aura MLS v4.2 100-hPa $H_2O$ over North America. The altitude is approximately 17 km. Each monthly histogram is normalized over mixing ratio such that, if one integrates colors (ppmv-1) over a range of mixing ratio, the result is a dimensionless probability that is normalized to one when integrating over all colors for that month. Thin vertical lines mark year boundaries, and gray-shaded areas denote July-August. Data are from the eastern CONUS through northern Mexico (see Figure 2 insert), a study box empirically chosen to enclose the highest outliers associated with

the North American Monsoon in the 10+ year MLS 100-hPa water vapor climatology (after Schwartz et al., 2013).

2) "The paper uses three vertical coordinates and doesn't tie them together very well. The introduction focuses on potential temperatures, everything pertaining to Figures 1-4 is discussed and shown in pressure coordinates, then Figures 5-7 are presented in altitude coordinates. Are the profiles in Figures 4-7 entirely in the lowermost stratosphere or do they extend into the overworld (or upper troposphere for that matter)? Profiles in Figures 4 extend from 150 to 50 hPa but in Figure 5a (same profiles) span 15.5 to 18.5 km. Are the axis ranges of these two Figures the same in terms of vertical span? I understand the need to discuss stratospheric layers in terms of potential temperature (introduction), but why can't everything else be presented uniformly using either pressure or altitude coordinates (or both together)? It would make the discussion and Figures much more intelligible."

Author's response to 2: To be consistent with previous literature on OT, the vertical coordinate of choice is altitude. We will change Figure 4 vertical coordinates to Altitude (left axis) and approximate Potential Temperature (right axis). Figures 1, 2 and 3 use the standard MLS 100-hPa product but we will modify each caption to 100 hPa (approximately 17 km altitude). The profiles in figures 4-7 are mostly overworld stratosphere, with some lowermost stratosphere at the bottom of the profile (no upper tropospheric data are shown). This will be made clear by the revised figure 4: altitudes 14.5 to 20 km and potential temperature from 365 K to 480 K.

Figure 4. Map and profiles of aircraft and satellite water vapor on 8 August 2013 over California (number 1 shown in dark blue) and Texas (number 2 shown in green). (a) Map of ER-2 aircraft flight track (solid colored trace) and nearly coincident Aura MLS geolocations (asterisks and lines). (b) ER-2 aircraft pressure profiles (solid colored trace) color-coded by dives and MLS times (horizontal lines). (c) Vertical profiles of water vapor from JLH Mark2 (dots), in situ with MLS averaging kernel (asterisks and lines), and MLS (circles and lines)

3) [1] "Though the phrase 'enhanced water vapor' appears in the paper's title and is used frequently throughout the paper, it is never defined for the aircraft measurements. [2] Presumably there is a mixing ratio threshold used to identify air parcels with enhanced water vapor? The abstract may indirectly imply, likely incorrectly, that "enhanced" mixing ratios measured from aircraft are those >10 ppmv. [3] Adding to the mystery are the blue markers in panels (a) of Figures 5-7 that represent the "enhanced H2O region" but range well below 5 ppmv. What exactly is the threshold for "enhanced" water vapor measured from the aircraft? Why are there mixing ratios <5 ppmv in regions of enhanced water vapor? This is quite confusing and requires some important clarification in the paper."

Author's response to 3: We agree with the referee that 'enhanced water vapor' should be presented more clearly. [1] A definition, added to the paper at line 84 (new lines 91-92 in revised manuscript), is: "Here we define 'enhanced water vapor' as a mixing ratio that is greater than two standard deviations above the mean in situ measurement." This is the threshold for 'enhanced water vapor' as measured from the aircraft. [2] In the revised paper, we will present a statistical analysis of the aircraft data, showing the value of mean plus 2 sigma, the threshold for enhanced water, that varies with pressure (new figure in preparation). In the overworld stratosphere, water vapor mixing ratios > 10 ppmv are very rare and thus "enhanced." [3] The referee is correct that panels (a) of Figures 5-7 are confusing. We will modify panels (a) so that enhanced water vapor is shown in a different color. We consider only the extreme elevated mixing ratios (mean + 2 sigma) to be "enhanced" water vapor.

4) [1] "Is it appropriate to try to combine information from MLS and aircraft-based detections of 'enhanced' water vapor in this paper when their thresholds are probably very different? Figure 3 shows only 13 instances where MLS retrievals at 100 hPa during Jul-Aug 2013 exceeded 8 ppmv. [2] Is this the threshold for MLS-detected "enhanced" water vapor? [3] Given the greatly different vertical and horizontal resolution of the MLS and aircraft measurements, how can the MLS findings be integrated with the

aircraft-based results that follow? I don't think they can. [4] What does a MLS retrieval of 8 ppmv with a 3-km averaging kernel width translate to for the in situ aircraft measurements? [5] Figures 1-3 contribute to only one conclusion drawn in this paper: that MLS retrievals at 100 hPa over the NAM region during Jul-Aug rarely exceed 8 ppmv. In my opinion that is a secondary (and already published) conclusion compared to the dominant conclusion of this paper that enhanced stratospheric water vapor measured over the NAM region during Jul-Aug 2013 can be traced back to convection-induced overshooting cloud tops."

Author's response to 4: To respond to the referee's comments, we will add information to the paper about comparing MLS and aircraft-based detections. This subject did not receive adequate description in the submitted manuscript, and is addressed below. 1) Rodgers and Connor (2003) mathematically described how to rigorously compare satellite remotely-sensed data with a different dataset (e.g., a high-resolution in situ dataset). Following from this article, the MLS data may be compared to the aircraft data once the MLS averaging kernel (observation operator) is applied to the in situ data. 2) The threshold for MLS-detected 'enhanced water vapor' is set at 8 ppmv, same as Schwartz et al. (2013), to exclude the larger population of measurements at 6 to 8 ppmv water vapor that may have other sources. 3) By using the MLS averaging kernel, we can address both the satellite data and the aircraft in situ data. 4) Figure 4 shows that an MLS retrieval of 7 ppmv with a 3-km averaging kernel width translates to 11 ppmv for in situ aircraft measurements. 5) Each summer has different meteorology, and we wish to use a decade of MLS measurements to place 2013 in context. There are three major results from the MLS figures that we want to retain in the paper, that summer 2013 had fewer extreme events than the previous three years (Figure 1), summer 2013 was drier on average than the previous nine summers, 2004-2012 (Figure 2), and the estimated frequency of $H_2O$ > 8ppmv was 0.9 percent (see below). The author wishes to keep the MLS figures in the paper to demonstrate these points. Please note that Schwartz et al. (2013, Geophys. Res. Lett.) did not show data from the year 2013.

Referee's Editorial Comments:

Line 58: "please give the lowermost stratosphere a rough lower limit (in potential temperature) otherwise this implies it extends down to the surface. "Commonly" suggests these mechanisms are highly probably pathways for tropospheric water vapor to reach the stratosphere, which they are not." Author's response: We have modified the underlined text at line 58: "In contrast to water entry into the overworld stratosphere, water transport from the troposphere to the lowermost stratosphere (350 K < < 380 K over summer CONUS) may occur through several different pathways. Poleward of the subtropical jet, water may be transported into the lowermost stratosphere through isentropic troposphere-stratosphere exchange (Holton et al., 1995) or through convective overshoot of the local tropopause (Dessler et al., 2007; Hanisco et al., 2007). Isentropic transport from the tropics is the dominant pathway for water into the lowermost stratosphere, with evidence from the seasonal cycle of water (e.g., Flury et al., 2013)."

Flury, T., Wu, D. L., and Read, W. G.: Variability in the speed of the Brewer-Dobson circulation as observed by Aura/MLS, Atmos. Chem. Phys., 13, 4563–4575, www.atmos-chem-phys.net/13/4563/2013/, doi:10.5194/acp-13-4563-2013, 2013.

Descent of middle stratospheric air into the lowermost stratosphere plays a smaller role than isentropic transport, as evidenced by relatively high water mixing ratios and low ozone mixing ratios of the lowermost stratosphere.

Line 62: "Please make it clear that ice (not elevated water) is transported into the stratosphere where it sublimates and produces "enhanced" water vapor mixing ratios. Author's response: we have modified the underlined text at line 62: "Case studies have reported extreme events in which ice is transported to the overworld stratosphere and subsequently sublimates, but the amount of ice that is irreversibly injected into the stratosphere is poorly known."

Line 63: "Here and elsewhere "stratospheric overworld" (defined in Line 45) has now become the "overworld stratosphere"." Author's response: Consistent with other publications, we will change the wording in line 45 and elsewhere from "stratospheric overworld" to "overworld stratosphere."

Line 66: "Ice does not "bypass" the cold trap, it is unaffected by it." Author's response: The referee is correct, we have modified the underlined text at line 66: "Ice injected directly into the stratosphere is unaffected by the cold trap in the vicinity of the tropopause (Ravishankara, 2012)."

Line 67: "Suggested paragraph break before "Paraphrasing"." Author's response: We inserted a paragraph break in line 67 with a new topic sentence: "The subject of this paper is the role of convective overshooting tops in enhancing stratospheric water."

Line 81: "are limited by their horizontal and vertical resolution in detecting fine-scale three-dimensional variations in water vapor ..." Author's response: the reviewer has a good suggestion. We will change this sentence to: "are limited by their horizontal and vertical resolution in detecting fine-scale three-dimensional variations in water vapor ..."

Line 101: " 'Instruments on the NASA ER-2' – which instruments and what was measured that allows you to conclude that 'the aircraft intercepted convectively-influenced air'?" Author's response: Only the hygrometers onboard the aircraft measured a tropospheric signature (e.g., enhanced water vapor). Other tracers did not show a tropospheric signature because lofted ice transported a disproportionate amount of H2O relative to gas-phase tracers. We have modified the text at line 101 to: "Enhanced water vapor measured in situ by both the JLH Mark2 instrument and the Harvard Water Vapor instrument (J. B. Smith, pers. comm.) on the NASA ER-2 aircraft indicated that the aircraft intercepted convectively-influenced air."

Line 107: " 'measures daily global atmospheric profiles' sounds like one gigantic profile is measured each day. How about 'measures ∼3500 profiles around the globe each day of atmospheric species ...' " Author's response: The reviewer has a good suggestion, we will change this sentence to "measures ∼3500 profiles each day of water

vapor and other atmospheric species (Livesey et al., 2016)."

Livesey, N. J., Read, W. G., Wagner, P. A., Froidevaux, L., Lambert, A., Manney, G. L., Millan Valle, L. F., Pumphrey, H. C., Santee, M. L., Schwartz, M. J., Wang, S., Fuller, R. A., Jarnot, R. F., Knosp, B. W., and Martinez, E.: Earth Observing System (EOS) Aura Microwave Limb Sounder (MLS) Version 4.2x Level 2 data quality and description document, Tech. Rep. JPL D-33509 Rev. B, Version 4.2x-2.0, Jet Propulsion Laboratory, California Institute of Technology, Pasadena, CA, available online at: http://mls.jpl.nasa.gov/data/datadocs.php (last access: May 9, 2016), 2016.

Line 118: "Aren't 'the decadal histogram' and 'the previous multi-year MLS record' the same thing in Figure 2? Don't you want to compare and contrast the histogram of 2013 with the histogram of 10-year mean values? Figure 2 would be a great place to visually show (as a vertical line) the threshold for 'enhanced' MLS water vapor." Author's response: Yes, the referee is correct, the 'decadal histogram' and the 'previous multi-year MLS record' are the same. The purpose of Figure 2 is to compare the histogram of July-August 2013 with the histogram of 9-summer mean values (also July-August). We have reworded this sentence in line 118 to: "Figure 2 shows that the July-August 2013 CONUS lower stratosphere was drier than the previous nine-summer MLS record (2004 to 2012)."

We have also added a vertical line to Figure 2 for the threshold for 'enhanced' MLS water vapor (8 ppmv). See attached Figure 2.

Figure 2. Distribution of Aura MLS v4.2 100-hPa H2O over CONUS (blue shaded box in insert), corresponding to approximately 17 km. The two histograms for July-August 2013 (blue asterisks and trace) and the previous nine-summer MLS record, July-August 2004 through 2012 (red circles and trace) indicates that 2013 was drier than average. The 8-ppmv threshold for "enhanced" water vapor is shown by the thick black line.

Line 120: "You have the histograms so why say 'rare' when you can be quantitative?" Author's response: We have added a sentence at line 126 stating: "From the MLS

histogram, the frequency of 100-hPa $H_2O$ > 8ppmv was 0.9% of the observations in July-August 2013 in the blue shaded box." We also modified the sentence at line 263 in the final paragraph: "The fraction of Aura MLS observations in the same time period with H2O greater than 8 ppmv is only 0.9%."

Line 122: "At least 3 of the white circles in Figure 3 are near the west coast of Mexico, not Central America." Author's response: We have changed the line 122 text to "water greater than 8 ppmv was measured only nine times over North America (in the blue shaded box), three times near the west coast of Mexico, and once over the Caribbean Sea."

Line 146: "What is the spatial resolution (horizontal and vertical) of the convective storm information used to link 'enhanced' water vapor to overshooting tops? Later (Line 503) you say that the OT data are 'high resolution' but the spatial resolution is never described." Author's response: The horizontal spatial resolution of the OT product is dependent on the underlying satellite imagery resolution, i.e., the size of the GOES IR pixel at any given spot.Âǎ The size goes as you move further away from the subsatellite point.Âǎ At subsatellite the pixel size is 4 km.Âǎ At the junction between GOES East and GOES West, the pixel size is about 7 km in Montana, probably 6+ km in Mexico.

When the referee asks about the spatial resolution in the vertical dimension of the convective storm information, perhaps a more relevant question is: what is the accuracy of the OT altitude? This has been addressed in Griffin et al. (JAMC, 2016), who report that 75% of OT height retrievals are within 0.5 km of CloudSat OT height.

We have added text to line 140: "For a description of the method, the reader is directed to Bedka et al. (2010). The horizontal spatial resolution of the OT product is dependent on the underlying satellite imagery resolution, i.e., the size of the GOES IR pixel, which is 7 km or less over the CONUS."

And text to line 146: "Griffin et al. (2016) finds that 75% of OT height retrievals are within 0.5 km of CloudSat OT height, so we conservatively estimate the accuracy of

the OT altitude to 0.5 km."

Line 156: "What was the time step interval for back trajectory calculations?" Author's response: We added the following text to line 155: "..., and the trajectory time step interval was one hour."

Line 158: "How were these tolerances chosen? Do they have any relationship to the spatial resolution (horizontal and vertical) of the convective storm information?" Author's response: We added the following text to the end of the paragraph (line 159): "These tolerances were chosen primarily due to the resolution of the NCEP meteorology used to run the trajectories (1 deg x 1 deg) and also based on personal communication with Leonard Pfister."

Line 164: You already wrote about initializations (Line 153). This statement is again repeated in Line 180. Author's response: At line 164, we have replaced "For each of these ER-2 flights, seven-day back trajectory analyses are initialized at locations and times of enhanced water vapor along the ER-2 flight track. These analyses are combined with overshooting cloud top data from the SEAC4RS OT data product." With "For each of these ER-2 flights, the back trajectories are presented along with the intersection of coincident OT." We also deleted the sentence at line 180, and moved the rest of the paragraph to after "Analysis of the 8 August 2013 case is shown in Figure 5."

Line 169: It would be beneficial to include the lats/lons of these sites. Author's response: we have modified the text to: "Palmdale, California (34.6 N, 118.1 W), to Ellington Field, Houston, Texas (29.6 N, 95.2 W)."

Line 173: Same comment as for Line 58. Author's response: we have changed the text to: "(350 K < < 380 K)"

Line 182: This information is already in the Figure Captions. Author's response: these sentences are needed to help explain the plots. We slightly modified the text to: "Anal-

ysis of the 8 August 2013 case is shown in Figure 5. For clarity only some example trajectories (a subset of our analysis) are shown. These are displayed as thin blue traces in panels (b) and (c). The initial water vapor mixing ratios of the example trajectories are shown as red squares in panels (a). The intersections of the example trajectories with coincident OT are shown as red squares in panels (b) and (c). All overshooting convective tops within +/-3 hours of the red squares are shown by green symbols in panels (b) and (c)."

Line 184: In Figures 5-7, panels (b) and (c), the green markers show "near coincidences" (within the tolerances listed in Line 158) between back trajectories and OTs, while red squares show "coincidences". What are the criteria for "coincidences"? Author's response: We thank the referee for catching this typo. The description in the original text doesn't quite match what's on the figure.Âă The red symbols are where there were coincidences for the specific example trajectories plotted in light blue.Âă The green symbols show all overshooting convective locations within +/- 3 hours of the red points, not related to where any of the trajectories went (see modified text above for line 182).Âă

We changed the text in Figure captions 5, 6 and 7 to remove "nearly coincident" since the only coincidence for green markers is in time. That doesn't really qualify as nearly coincident. The reason to show all the green symbols is to give an indication of how robust the coincidences are. For instance, the mass of green points north of Texas on Aug. 8 indicates there were a lot of overshoots there and the coincidences should be robust for the trajectories that went through that region.ÂăThe two coincidences in Arizona are among only a small cluster of overshoots and so this is not as robust a coincidence.

The criteria for coincidence (red markers) is, as described in Section 3.2: "+/-0.25 degrees latitude and longitude, +/-3 hours, +/-0.5 km in altitude."

Line 191: In Figure 5b there are many green markers in western Mexico, so why is this

location not listed? Author's response: As described above (line 184), green markers are not coincident. Only the red markers are coincident in space and time, so we only describe the locations of the red markers.

Line 198: In Figure 5d there are many red markers depicting transit times >6 days, but here you claim domination by transit times of "two to five days". Author's response: in line 198, we changed "two to five days earlier than" to "within seven days prior to".

Line 205: TX, OK and AR are in the "South Central" U.S. Author's response: in line 205 we have replaced "Central" with "South Central".

Line 216: Define MCC as mesoscale convective complex Author's response: in line 216 we have replaced "MCC" with "mesoscale convective complex".

Line 227: Earlier you stated that back trajectories were initialized for every measurement of enhanced water vapor. Now, "Example" infers that this was done only for a subset. Did "all" (Line 228) of the back trajectories connect to OTs? Really? All? Author's response: The referee has an excellent point here. Not all of the back trajectories connect to OTs, although a majority do. This will be made more quantitative in the revision. The trajectories (initialized for every measurement of enhanced water vapor) are a way to show that the enhanced water comes from OT over CONUS. We will delete the old text "Example air parcel back-trajectories were initialized at the locations and time of enhanced water. All of the back-trajectories connect these air parcels to convective OT one to seven days prior to aircraft intercept." And add the following: "Back trajectories initialized on the flight track where enhanced water vapor was measured connect the sampled air parcels to convective OT within seven days prior to the flight."

Line 230: Figure 7b shows there were influences from storm systems in South Central Canada as well. Author's response: We have added to line 231: "and South Central Canada."

Line 235: "leaving behind" should be "producing" Author's response: we have changed

the text at line 235 to: "... producing water vapor mixing ratios elevated up to 15 ppmv above background levels."

Line 238: "propel water" gives the wrong impression while "loft ice" is more accurate. Author's response: we have changed "propel water" to "loft ice".

Line 239: "the enriched delta-D isotopic signature" needs a bit more explanation, including a mention of the isotope itself, HDO. Author's response: We modified the text as follows at line 239: "Further evidence of ice is provided by water isotopologues. Evaporation and condensation are fractionating processes for isotopologues, especially HDO relative to $H_2O$ (e.g., Craig, 1961; Dansgaard, 1964). Condensation preferentially concentrates the heavier HDO isotopologue, so lofted ice is relatively enriched in $HDO/H_2O$ compared to gas phase (e.g., Webster and Heymsfield, 2003, and references therein). Ice sublimation is supported by the enriched $HDO/H_2O$ isotopic signature observed by the ACE satellite over summertime North America (Randel et al., 2010)."

New references: Craig, H.: Isotopic Variations in Meteoric Waters, Science, 133, 1702-3, doi: 10.1126/science.133.3465.1702, 1961. Dansgaard, W.: Stable isotopes in precipitation, Tellus, 16, 436-68, 1964.

Line 246: "The water is almost certainly injected in the ice phase" needs support. I suggest you combined this paragraph with all or some of the previous paragraph that provides such support. Author's response: we have merged this paragraph with the previous paragraph.

Line 252: "from a long (200 km) path through the atmosphere" also needs to include information about the vertical resolution. The qualitative statement "may be enhanced even more" is the basis for my argument that the MLS and aircraft results cannot be meaningfully integrated together in this paper. Author's response: The revised Figure 4 (see above) demonstrates that we can compare satellite and in situ measurements and, furthermore, map the aircraft to MLS resolution through averaging kernels. We

will describe Figure 4 better in the text.

Line 254: what percent of the 10 years of MLS observations show enhanced water? Author's response: we have modified the sentence at line 264 (for better flow): "The fraction of Aura MLS observations at 100 hPa with $H_2O$ greater than 8 ppmv is only 0.9% for July-August 2013. Schwartz et al. (2013) reports that, for the nine-year record 2004-2012, July and August had 1.4% and 3.2% of observations exceed 8 ppmv, respectively."

Line 261: Which "monsoon region"? Author's response: Randel et al. (2015) addresses the North American Monsoon (NAM) region. We have changed the text in line 261 to: "NAM region."

Figure 2: Caption: "ten year mean for 2004 through 2013." Author's response: We decided to compare 2013 with the previous nine-summer record, 2004 through 2012. Figure 2 has been updated, and the caption changed to: "Distribution of Aura MLS v4.2 100-hPa $H_2O$ over CONUS (blue shaded box in insert), corresponding to approximately 17 km. The two histograms for July-August 2013 (blue asterisks and trace) and the previous nine-summer MLS record, July-August 2004 through 2012 (red circles and trace) indicates that 2013 was drier than average. The 8-ppmv threshold for "enhanced" water vapor is shown by the thick black line."

Figure 3: Caption: "Average Aura MLS 100-hPa" Author's response: we will change the caption to "Two-month mean map of Aura MLS v4.2 100-hPa..."

Figure 4: "[1] Colored markers are not "retrievals" from the aircraft, they are the actual aircraft measurements. [2] Why do multiple aircraft profiles produce only one profile convolved with the MLS averaging kernels? [3] Without strongly magnifying this Figure I can't tell the difference between the MLS profiles and the convolved aircraft profile. [4] I suggest omitting the convolved aircraft profile in each panel. It does shows that the averaging kernels smooth the aircraft profiles, but isn't that exactly what one would expect anyway? [5] Also, the black asterisks showing MLS retrieval locations on flight

track maps are difficult to distinguish from black map lines. Perhaps larger gray symbols "x" or "+" would stand out more? [6] And the "line" mentioned in the caption, is this the horizontal line in each "Flight Altitude" (should be "Pressure") vs UTC hour panel indicating the time range of measurements shown in the profiles? [7] This Figure would be an ideal place to visually show (as a vertical line in each profile panel) the threshold for aircraft "enhanced" water vapor." Author's response on Figure 4: These are great points by the referee. 1) We will change the figure 4 caption from "water vapor retrievals from aircraft (color), aircraft with MLS averaging kernel (asterisk and lines)" to "JLH Mark2 in situ water vapor data (dots), in situ with MLS averaging kernel (asterisks and lines), . . ." 2) In each Figure 4 panel, the pair of aircraft profiles (descending and ascending) are combined to produce one profile convolved with the MLS averaging kernels. 3) Figure 4 has been redone (see above). 4) We wish to keep the convolved aircraft profile to demonstrate that 7 ppmv MLS corresponds to approximately 11 ppmv in situ aircraft data. 5) Figure 4 MLS symbols are now in color (see above). 6) The Pressure vs UTC hour panel is now altitude vs time, properly labeled "Altitude", with new caption "MLS times (horizontal lines)". 7) We will add text to describe the threshold for aircraft "enhanced" water mixing ratios (still in work).

Figure 5: These comments apply to each of Figures 5-7: In panels (a), the captions claim the blue markers denote "enhanced water measurements" (which range below 5 ppmv) while in the panel (a) legends the blue markers are said to represent the "Enhanced H2O region", which must be something quite different from "enhanced water vapor" measurements. This distinction needs to be clarified in the paper by defining exactly what is meant by the terms "enhanced water measurements" and "enhanced H2O region". Author's response: The referee has a good point, we will remake the Figures 5-7 to clearly delineate the enhanced water mixing ratios (still in work).

[Figure]

**Fig. 1.** Figure 1. Time series of Aura MLS v4.2 100-hPa H2O over North America - see text for full caption.

[Figure]

**Fig. 2.** Figure 4. (a) Map of ER-2 track and MLS geolocations. (b) Pressure profile of ER-2 aircraft. (c) comparison of JLH Mark2 H2O with MLS - see text for full caption.See text for full caption.

- Y-axis: MLS Observations in July-August, 2013 (0 to 250)
- X-axis: 100-hPa H $_2$O in North American Box (ppmv) (2 to 10)
- Legend: 2013 (blue); 2004-2012 average (red)

Inset map with axes: 60, 40, 20, 0 and -150, -120, -90, -60

**Fig. 3.** Figure 2. Histogram of MLS H2O - see text for full caption.

---

## Author Response (AR1)

Authors' response to anonymous referee #1 on "Enhanced Stratospheric Water Vapor over the Summertime Continental United States and the Role of Overshooting Convection" by R. L. Herman et al., ACP-2016-1065

We would like to thank the referee #1 for detailed review, and for insightful and constructive comments. This response replaces the previous authors' comment (in January 2017). This response addresses all of the referee's individual points below:

**Major Comment**

**Referee's Major Comment:**

"In my opinion this paper could easily stand alone if all Figures and in-depth discussions of Aura MLS stratospheric water vapor measurements were omitted. The vast majority of scientific conclusions can be made without the broader picture provided by the MLS data. The one significant contribution of MLS data to this paper is to show the low frequency of occurrence of anomalously high stratospheric water vapor mixing ratios over the North American monsoon region. However, this conclusion, based on MLS water vapor data, was already published by Schwartz et al. (2013, Geophy. Res. Lett.). There are also some difficulties translating between the aircraft and MLS results because of the presumably different mixing ratio thresholds used to identify "enhanced" water vapor for each. Intuitively the MLS threshold needs to be lower because of the much coarser vertical and horizontal resolution of MLS retrievals. See my comments below regarding this issue."

"I will leave it up to the authors if they want to retain or omit the MLS data in their paper. I don't think its presence detracts from the main objectives of this paper, but I also think it doesn't contribute much to them."

**Authors' Response to Major Comment:**

The Aura MLS stratospheric water vapor measurements place the aircraft field mission data into regional and decadal perspective. We have improved the text and figures (as described below) to tie MLS and the aircraft results together. MLS helps us address the question from Toon et al (2016): "Do deep convective cloud systems locally inject water vapor and other chemicals into the overworld stratosphere over the CONUS?"

We do agree with the referee that our old Figure 1 (MLS 2004-2013 time series) is similar to a previously published figure (MLS 2004-2012 time series from Schwartz et al., 2013), and will be removed from the revised paper.

**Authors' Changes in Manuscript in response to Major Comment:**

The old Figure 1 (MLS decadal time series) has been replaced by a histogram of MLS water (new Figure 4). Also, in our revised paper we show that both MLS water at 100 hPa and aircraft water are enhanced significantly above background with revised plots of both aircraft and MLS data (Figures 1, 4 and 6).

**Minor Comments Referee's Minor Comment 1)**

Figure 1 and Caption. "'Each monthly histogram is normalized to unity over mixing Ratio' needs further explanation. I went back to the Schwartz et al. (2013) paper for clarification and found the exact same statement. For me, to normalize each monthly histogram one would divide the population of each mixing ratio bin by the entire population or the population of the bin containing the mode or mean of the distribution. Assuming a somewhat Gaussian distribution, the normalization the mode or mean bin population would produce numbers near one and zero for the most and least populated bins (most and least probable in a PDF), respectively. This is how I interpret Figure 1 even though I don't understand "normalized to unity over mixing ratio" and the units on the color bar are ppmv^-1. What am I missing here?"

**Author's response and change to manuscript for referee's comment 1) We have removed the old Figure 1 because it is unnecessary.**

**Referee's Comment on Figure 1:**

Figure 1 comment: "Also for Figure 1, the vertical lines are not "dashed" and the gray shading is too light, at least for my eyes."

**Author's response and change to manuscript for Figure 1:**

We have removed the old Figure 1 of MLS data because it is redundant with other figures.

**Referee's Comment 2)**

"The paper uses three vertical coordinates and doesn't tie them together very well. The introduction focuses on potential temperatures, everything pertaining to Figures 1-4 is discussed and shown in pressure coordinates, then Figures 5-7 are presented in altitude coordinates. Are the profiles in Figures 4-7 entirely in the lowermost stratosphere or do they extend into the overworld (or upper troposphere for that matter)? Profiles in Figures 4 extend from 150 to 50 hPa but in Figure 5a (same profiles) span 15.5 to 18.5 km. Are the axis ranges of these two Figures the same in terms of vertical span? I understand the need to discuss stratospheric layers in terms of potential temperature (introduction), but why can't everything else be presented uniformly using either pressure or altitude coordinates (or both together)? It would make the discussion and Figures much more intelligible."

**Author's response to comment 2:**

To be consistent with previous literature on OT, the vertical coordinate of choice is altitude. We will change Figure 6 vertical coordinates to Altitude (left axis) and approximate Potential Temperature (right axis). Figures 4, 5, and 6 use the standard MLS 100-hPa product but we will modify each caption to 100 hPa (approximately 17 km altitude). The profiles in old figures 4-7 (revised figures 6-9) are mostly overworld stratosphere, with some lowermost stratosphere at the bottom of the profile (no upper tropospheric data are shown). This will be made clear by the revised y-axis of Figures 6, 7a, 8a, 9a.

Authors' Changes in Manuscript to comment 2:

New text in Abstract: "...were measured by the JPL Laser Hygrometer (JLH Mark2) at altitudes between 16.0 and 17.5 km (potential temperatures of approximately 380 K to 410 K)." New text in new Figure 4 (old Figure 2) caption: "**Figure 4.** Distribution of Aura MLS v4.2 100hPa H2O over CONUS (blue shaded box in insert), corresponding to approximately 17 km..." New text in new Figure 5 (old Figure 3) caption: "**Figure 5.** Two-month mean map of Aura MLS v4.2 100-hPa H2O (color scale), corresponding to approximately 17 km altitude, with superimposed MERRA horizontal winds (arrows) for July-August 2013 during the SEAC4RS time period."

New Figure 6 and new Figure 6 caption:

**Figure 6.** Map and profiles of aircraft and satellite water vapor on 8 August 2013 over California (number 1 shown in dark blue) and Texas (number 2 shown in green). (a) Map of ER-2 aircraft flight track (solid colored trace) and nearly coincident Aura MLS geolocations (asterisks and lines). (b) ER-2 aircraft pressure profiles (solid colored trace) color-coded by dives and MLS times (horizontal lines). (c) Vertical profiles of water vapor from JLH Mark2 (dots), *in situ* with MLS averaging kernel (asterisks and lines), and MLS (circles and lines).

New revised y-axis of Figures 6, 7a, 8a, 9a (altitude and potential temperature) demonstrates that the plotted measurements are from the lowermost stratosphere and overworld stratosphere, not the troposphere. We have also replaced "UTLS" with "lower stratosphere" where appropriate in the text.

**Referee's Comment 3):**

[1] "Though the phrase 'enhanced water vapor' appears in the paper's title and is used frequently throughout the paper, it is never defined for the aircraft measurements. [2] Presumably there is a mixing ratio threshold used to identify air parcels with enhanced water vapor? The abstract may indirectly imply, likely incorrectly, that "enhanced" mixing ratios measured from aircraft are those >10 ppmv. [3] Adding to the mystery are the blue markers in panels (a) of Figures 5-7 that represent the "enhanced H2O region" but range well below 5 ppmv. What exactly is the threshold for "enhanced" water vapor measured from the aircraft? Why are there mixing ratios <5 ppmv in regions of enhanced water vapor? This is quite confusing and requires some important clarification in the paper."

**Author's response to comment 3:**

We agree with the referee that 'enhanced water vapor' should be presented more clearly. [1] The phrase 'enhanced water vapor' will be defined (see below in author's changes). [2] A mixing ratio threshold is identified as the mean plus 2 st. dev., which is 9.7 ppmv at 380-400 K, and 6.6 ppmv at 400-420 K (see below in author's changes).

[3] The referee is correct that panels (a) of Figures 5-7 are confusing, so these will be changed.

Author's changes in manuscript in response to comment 3:

[1] A definition, added to the paper at line 84 (new lines 107-108 in revised manuscript), is:

"Here we define 'enhanced water vapor' as mixing ratios greater than two standard deviations above the mean in situ measurement." This is the threshold for 'enhanced water vapor' as measured from the aircraft.

[2] In the revised paper, we present a statistical analysis of the aircraft data, characterizing the mean, standard deviation and distribution of water vapor. The threshold for enhanced water is the campaign-wide mean plus 2 standard deviations. New text added at lines 108-110 in revised manuscript:

"For the overworld stratosphere in all 23 SEAC4RS flights, mean  $H_2O$  for is 6.7±1.5 ppmv at 380-400 K (Figure 2), and 5.0±0.8 ppmv at 400-420 K (Figure 3). Thus the threshold for enhanced water vapor is 9.7 ppmv at 380-400 K, and 6.6 ppmv at 400-420 K."

In the overworld stratosphere (potential temperature > 380 K), water vapor mixing ratios > 10 ppmv are unusual and thus "enhanced." I should emphasize that the overworld stratosphere is typically drier than the lowermost stratosphere (350-380 K).

[3] The response to this comment is the same as the response to Figure 5 (see below in Referee's Editorial Comments). We agree with the referee that this discussion is confusing, and have made changes to the manuscript. As described in lines 108-109, the threshold for 'enhanced water vapor' is mean + 2 sigma (from the combination of 23 aircraft flights) binned in two layers: 380-400 K and 400-420 K potential temperature. We observe these 'enhanced water vapor' events at 380-410 K (see Figure 1), so we call this layer the 'enhanced water region.' New text added at lines 112-113 in the revised manuscript:

"We define the 'enhanced water region' as the layer of the overworld stratosphere where these events have been observed, 380-410 K potential temperature corresponding to 16-17.5 km altitude."

The blue points in (old) Figure 5 were confusing, so we removed them (see new Figures 7,8,9).

**Referee's Comment 4)**

[1] "Is it appropriate to try to combine information from MLS and aircraft-based detections of water vapor in this paper when their thresholds are probably very different? Figure 3 shows only 13 instances where MLS retrievals at 100 hPa during Jul-Aug 2013 exceeded 8 ppmv. [2] Is this the threshold for MLS-detected "enhanced" water vapor? [3] Given the greatly different vertical and horizontal resolution of the MLS and aircraft measurements, how can the MLS findings be integrated with the aircraft-based results that follow? I don't think they can. [4] What does a MLS retrieval of 8 ppmv with a 3-km averaging kernel width translate to for the in situ aircraft measurements? [5] Figures 1-3 contribute to only one conclusion drawn in this paper: that MLS retrievals at 100 hPa over the NAM region during Jul-Aug rarely exceed 8 ppmv. In my opinion that is a secondary (and already published) conclusion compared to the dominant conclusion of this paper that enhanced stratospheric water vapor measured over the NAM region during Jul-Aug 2013 can be traced back to convection-induced overshooting cloud tops."

**Author's response to comment 4)**

[1] We believe that it is appropriate to report information from MLS and aircraft because both satellite and aircraft show signals above background levels of water vapor in summer 2013, as shown in the revised Figure 6. We are not combining the MLS and aircraft datasets (although

Rodgers and Connor, 2003, showed mathematically how to do this). Instead, we are using both to describe the distribution of water vapor over the summer 2013 continental U.S. [2] Yes, the referee is correct, the threshold for MLS-detected 'enhanced water vapor' is set at 8 ppmv. This is the same threshold that Schwartz et al. (2013) used to exclude the larger population of measurements at 6 to 8 ppmv water vapor that may have other sources. [3] By using the histograms of overworld stratospheric water measured by aircraft (Figures 2 and 3) and MLS (Figure 4), we can conclude that these extreme events constitute only a percent or two of stratospheric observations. Furthermore, Figure 6 shows that both aircraft and MLS have significantly higher water vapor mixing ratios in regions with enhanced stratospheric water.

[4] We have calculated that an MLS retrieval of 8 ppmv with a 3-km averaging kernel width translates to 13 ppmv for *in situ* aircraft measurements. Three flights encountered greater than 13 ppmv in the overworld stratosphere, 12 Aug 2013, 27 Aug 2013 and 6 Sep 2013 (see Table 1).

[5] Each summer has different meteorology, and we wish to use a decade of MLS measurements to place 2013 in context. The Schwartz et al. (2013, Geophys. Res. Lett.) paper did not show data from the year 2013. There are three major results from the MLS figures that we want to retain in the paper: Summer 2013 was drier on average than the previous nine summers, 2004-2012 (Figure 4), the estimated frequency of  $H_2O > 8ppmv$  was 0.9 percent (see below), and MLS clearly retrieved enhanced water vapor during the SEAC4RS time period (Figure 5). The author wishes to keep the MLS figures in the paper to demonstrate these points.

**Author's changes to manuscript for comment 4)**

[1] revised Figure 6

[2] New text in Figure 4 caption: "The threshold for MLS-detected 'enhanced water vapor' (thick black vertical line) is set at 8 ppmv, same as Schwartz et al. (2013), to exclude the larger population of measurements at 6 to 8 ppmv water vapor that may have other sources."
[3] We have introduced new histograms of JLH Mark2 water vapor (Figures 2 and 3), and updated the histogram of MLS water vapor (Figure 4) to integrate the aircraft and satellite findings.

[4] no change to manuscript for this point.

[5] revised Figures 4, 5, 6 and updated text in the Conclusions.

Referee's Editorial Comments:

Referee's Comment on (old) Line 58:

"please give the lowermost stratosphere a rough lower limit (in potential temperature) otherwise this implies it extends down to the surface. "Commonly" suggests these mechanisms are highly probably pathways for tropospheric water vapor to reach the stratosphere, which they are not."

**Author's response to comment on (old) line 58:**

The lowermost stratosphere extends from the tropopause (approx. 350 K over summer CONUS) to 380 K potential temperature.

We are unclear what the referee meant by "highly probable mechanisms for tropospheric water vapor to reach the stratosphere" because there are only three major mechanisms for tropospheric water vapor to reach the stratosphere:

- a) Transport through the tropical tropopause layer (TTL) followed by isentropic transport,
- b) Mixing across the tropopause,
- c) Overshooting tops.

Descent of middle stratospheric air brings very dry air to the lowermost stratosphere. The relatively high water mixing ratios and low ozone mixing ratios of the lowermost stratosphere indicate that water is transported from the troposphere to the lowermost stratosphere. Isentropic transport from the tropics is the dominant pathway for water into the lowermost stratosphere, with evidence from the seasonal cycle of water (e.g., Flury et al., 2013). What we mean at (old) line 58 is that high water mixing ratios in the midlatitude stratosphere are most likely caused by either mechanisms a, b, and/or c.

**Author's change to manuscript in response to (old) line 58:**

We have modified the text at lines 58-63 in the revised manuscript:

"In contrast to water entry into the overworld stratosphere, water transport from the troposphere to the lowermost stratosphere (350 K <  $\theta$  < 380 K over summer CONUS) may occur through several different pathways. Poleward of the subtropical jet, water may be transported into the lowermost stratosphere through isentropic troposphere-stratosphere exchange (Holton et al., 1995) or through convective overshoot of the local tropopause (Dessler et al., 2007; Hanisco et al., 2007). Isentropic transport from the tropics is the dominant pathway for water into the lowermost stratosphere, with evidence from the seasonal cycle of water (e.g., Flury et al., 2013)."

Flury, T., Wu, D. L., and Read, W. G.: Variability in the speed of the Brewer-Dobson circulation as observed by Aura/MLS, *Atmos. Chem. Phys.*, *13*, 4563–4575, www.atmos-chem-phys.net/13/4563/2013/, doi:10.5194/acp-13-4563-2013, 2013.

Referee's Comment on (old) Line 62: "Please make it clear that ice (not elevated water) is transported into the stratosphere where it sublimates and produces "enhanced" water vapor mixing ratios.

Author's response to (old) Line 62: The referee is correct.

**Author's change to the manuscript for (old) line 62:**

We have modified the underlined text at lines 65-67 in the revised manuscript: "Case studies have reported extreme events in which ice is transported to the overworld stratosphere and subsequently sublimates, but the amount of ice that is irreversibly injected into the stratosphere is poorly known."

Referee's Comment on (old) Line 63: "Here and elsewhere "stratospheric overworld" (defined in Line 45) has now become the "overworld stratosphere"."

Author's response to (old) Line 63: Consistent with other publications, we will change the wording in line 45 and elsewhere from "stratospheric overworld" to "overworld stratosphere."

Author's change to the manuscript for (old) line 63:

Throughout the revised manuscript, we now consistently use "overworld stratosphere"

Referee's Comment on (old) line 66: "Ice does not "bypass" the cold trap, it is unaffected by it."

Author's response to (old) line 66: The referee is correct.

Author's change to manuscript for (old) line 66:

We have modified the underlined text at lines 69-70 in the revised manuscript: "Ice injected directly into the stratosphere is unaffected by the cold trap in the vicinity of the tropopause (Ravishankara, 2012)."

Referee's Comment on (old) line 67: "Suggested paragraph break before "Paraphrasing"."

Author's response to (old) line 67: This is a good suggestion.

Author's change to manuscript for (old) line 67: We inserted a paragraph break at new line 72 with a new topic sentence: "The subject of this paper is the role of convective overshooting tops in enhancing stratospheric water. Paraphrasing..."

Referee's Comment on (old) line 81: "are limited by their horizontal and vertical resolution in detecting fine-scale three-dimensional variations in water vapor ..."

Author's response to (old) line 81: the reviewer has a good suggestion.

Author's change to manuscript for (old) line 81: We have modified the text at lines 86-87 in the revised manuscript to: "Satellite measurements are limited by their horizontal and vertical resolution in detecting finescale three-dimensional variations in water vapor ..."

Referee's Comment on (old) line 101: "Instruments on the NASA ER-2' – which instruments and what was measured that allows you to conclude that 'the aircraft intercepted convectively-influenced air'?"

Author's response to (old) line 101: Only the hygrometers onboard the aircraft measured a convective signature (e.g., enhanced water vapor). Other tracers (including  $O_3$ ,  $CO_2$ , CO,  $CH_4$ ) did not show a convective signature. We infer that lofted ice transported a disproportionate amount of condensed  $H_2O$  relative to gas-phase tracers into the stratosphere.

Author's change to manuscript for (old) line 101:

We have modified the text at lines 117-120 in the revised manuscript to: "Enhanced water vapor measured *in situ* by both the JLH Mark2 instrument and the Harvard Water Vapor instrument (J. B. Smith, pers. comm.) on the NASA ER-2 aircraft indicated that the aircraft intercepted convectively-influenced air. Other tracers measured on the aircraft did not change significantly in these plumes."

Referee's Comment on line 107: "'measures daily global atmospheric profiles' sounds like one gigantic profile is measured each day. How about 'measures ~3500 profiles around the globe each day of atmospheric species ...'"

Author's response to line 107: The reviewer has a good suggestion.

Author's change to manuscript for (old) line 107: We have changed the text at line 125 in the revised manuscript to: "Aura MLS measures ~3500 profiles each day of water vapor and other atmospheric species (Livesey et al., 2016)."

Livesey, N. J., Read, W. G., Wagner, P. A., Froidevaux, L., Lambert, A., Manney, G. L., Millan Valle, L. F., Pumphrey, H. C., Santee, M. L., Schwartz, M. J., Wang, S., Fuller, R. A., Jarnot, R. F., Knosp, B. W., and Martinez, E.: Earth Observing System (EOS) Aura Microwave Limb Sounder (MLS) Version 4.2x Level 2 data quality and description document, Tech. Rep. JPL D-33509 Rev. B, Version 4.2x-2.0, Jet Propulsion Laboratory, California Institute of Technology, Pasadena, CA, available online at: http://mls.jpl.nasa.gov/data/datadocs.php (last access: May 9, 2016), 2016.

Referee's comment on line 118: "Aren't 'the decadal histogram' and 'the previous multi-year MLS record' the same thing in Figure 2? Don't you want to compare and contrast the histogram of 2013 with the histogram of 10-year mean values? Figure 2 would be a great place to visually show (as a vertical line) the threshold for 'enhanced' MLS water vapor."

**Author's response to line 118:**

Yes, the referee is correct, the 'decadal histogram' and the 'previous multi-year MLS record' are the same. The purpose of new Figure 4 (old Figure 2) is to compare the histogram of July-August 2013 with the histogram of 9-summer mean values (also July-August).

**Author's change to manuscript for (old) line 118:**

We have reworded this sentence at lines 135-137 in the revised manuscript: "The histogram of Aura MLS water vapor in Figure 4 indicates that the July-August 2013 CONUS lower stratosphere was drier than the previous nine-summer MLS record (2004 to 2012)."

We have also added a vertical line to Figure 4 for the threshold for 'enhanced' MLS water vapor (8 ppmv). See Figure 4 in the revised manuscript and the new caption below:

"Figure 4. Distribution of Aura MLS v4.2 100-hPa H2O over CONUS (blue shaded box in insert), corresponding to approximately 17 km altitude. The two histograms for July-August 2013 (blue asterisks and trace) and the previous nine-summer MLS record, July-August 2004 through 2012

(red circles and trace) indicates that 2013 was drier than average. The 8-ppmv threshold for "enhanced" water vapor is shown by the thick black line."

Referee's comment on line 120: "You have the histograms so why say 'rare' when you can be quantitative?"

Author's response on line 120: the referee has a good point here.

Author's change to manuscript for (old) line 120:

We have inserted a new sentence at line 138-139 in the revised manuscript:

"From the MLS histogram, the frequency of 100-hPa  $H_2O > 8$  ppmv was 0.9% of the observations in July-August 2013 in the blue shaded box."

We also modified the sentence at line 275 in the final paragraph:

"The fraction of Aura MLS observations at 100 hPa with  $H_2O$  greater than 8 ppmv is only 0.9% for July-August 2013."

Referee's comment on line 122: "At least 3 of the white circles in Figure 3 are near the west coast of Mexico, not Central America."

Author's response on line 122: The referee has a good point here about old Figure 3 (new Figure 5 in the revised manuscript).

Author's change to manuscript for (old) line 122:

We have changed the text at lines 140-142 in the revised manuscript to: "water greater than 8 ppmv was measured only nine times over North America (in the blue shaded box), three times near the west coast of Mexico, and once over the Caribbean Sea."

Referee's comment on line 146: "What is the spatial resolution (horizontal and vertical) of the convective storm information used to link 'enhanced' water vapor to overshooting tops? Later (Line 503) you say that the OT data are 'high resolution' but the spatial resolution is never described."

Author's response on line 146: The horizontal spatial resolution of the OT product is dependent on the underlying satellite imagery resolution, i.e., the size of the GOES IR pixel at any given spot. The size goes as you move further away from the subsatellite point. At subsatellite the pixel size is 4 km. At the junction between GOES East and GOES West, the pixel size is about 7 km in Montana, probably 6+ km in Mexico.

When the referee asks about the spatial resolution in the vertical dimension of the convective storm information, perhaps a more relevant question is: what is the accuracy of the OT altitude? This has been addressed in Griffin et al. (JAMC, 2016), who report that 75% of OT height retrievals are within 0.5 km of CloudSat OT height.

Author's changes to manuscript for (old) line 146:

We have added text to lines 160-162 in the revised manuscript:

"For a description of the method, the reader is directed to Bedka et al. (2010). The horizontal spatial resolution of the OT product is dependent on the underlying satellite imagery resolution, i.e., the size of the GOES IR pixel, which is 7 km or less over the CONUS."

And text to line 168-170 in the revised manuscript:

"Griffin et al. (2016) finds that 75% of OT height retrievals are within 0.5 km of CloudSat OT height, so we conservatively estimate the accuracy of the OT altitude to 0.5 km."

Referee's comment on line 156: "What was the time step interval for back trajectory calculations?"

Author's response and change to manuscript for (old) line 156: To answer this question, we added the following text to lines 207-208 in the revised manuscript: "..., and the trajectory time step interval was one hour."

Referee's comment on line 158: "How were these tolerances chosen? Do they have any relationship to the spatial resolution (horizontal and vertical) of the convective storm information?"

Author's response and change to manuscript for (old) line 158: To answer this question, we added the following text to the end of the paragraph (lines 211-213 in the revised manuscript): "These tolerances were chosen primarily due to the resolution of the NCEP meteorology used to run the trajectories (1 deg x 1 deg) and also based on personal communication with Leonard Pfister."

Referee's comment on line 164: You already wrote about initializations (Line 153). This statement is again repeated in Line 180.

Author's response on line 164: the referee has a good point and we will modify the text.

Author's change to manuscript for (old) line 164:

The new text (removing initialization text) at line 218-219 in the revised manuscript is: "For each of these ER-2 flights, the back trajectories are presented along with the intersection of coincident OT."

We also deleted the sentence at (old) line 180, and moved the rest of the paragraph to line 234 in the revised manuscript, after "Analysis of the 8 August 2013 case is shown in Figure 7."

Referee's comment on line 169: It would be beneficial to include the lats/lons of these sites.

Author's response and changes to the manuscript for (old) line 169: we have modified the text at line 222-223 in the revised manuscript to:

"Palmdale, California (34.6 °N, 118.1 °W), to Ellington Field, Houston, Texas (29.6 °N, 95.2 °W)."

Referee's comment on line 173: Same comment as for Line 58.

Author's response to line 173: we will give a lower limit to the lowermost stratosphere: 350 K potential temperature.

Author's change to manuscript for (old) line 173: At line 226-227 in the revised manuscript we have changed the text to: "(350 K <  $\theta$  < 380 K)"

Referee's comment on line 182: This information is already in the Figure Captions.

Author's response on line 182: the referee has a good point but we feel that some text is required here to guide the reader through the plots in new Figure 7 (old Figure 5).

**Author's change to manuscript for (old) line 182:**

Starting at line 234 in the revised manuscript, we slightly modified the text to: "Analysis of the 8 August 2013 case is shown in Figure 7. For clarity only some example trajectories (a subset of our analysis) are shown. These are displayed as thin blue traces in panels (b) and (c). The initial water vapor mixing ratios of the example trajectories are shown as red squares in panels (a). The intersections of the example trajectories with coincident OT are shown as red squares in panels (b) and (c). All overshooting convective tops within +/-3 hours of the red squares are shown by green symbols in panels (b) and (c)."

Referee's comment on line 184: In Figures 5-7, panels (b) and (c), the green markers show "near coincidences" (within the tolerances listed in Line 158) between back trajectories and OTs, while red squares show "coincidences". What are the criteria for "coincidences"?

Author's response on line 184: We thank the referee for catching this typo. The description in the original text doesn't quite match what's on the figure. The red symbols are where there were coincidences for the specific example trajectories plotted in light blue. The green symbols show all overshooting convective locations within +/- 3 hours of the red points, **not** related to where any of the trajectories went (see modified text above).

We changed the text in (old) Figure captions 5, 6 and 7 to remove "nearly coincident" since the only coincidence for green markers is in time. That doesn't really qualify as nearly coincident. The reason to show all the green symbols is to give an indication of how robust the coincidences are. For instance, the mass of green points north of Texas on Aug. 8 indicates there were a lot of overshoots there and the coincidences should be robust for the trajectories that went through that region. The two coincidences in Arizona are among only a small cluster of overshoots and so this is not as robust a coincidence.

The criteria for coincidence (red markers) is, as described in Section 3.2: "+/-0.25 degrees latitude and longitude, +/-3 hours, +/-0.5 km in altitude."

**Author's change to manuscript for (old) line 184:**

We removed "nearly coincident" from the captions for (old) Figures 5, 6 and 7 (new Figures 7, 8, 9 in the revised manuscript).

Referee's comment on line 191: In Figure 5b there are many green markers in western Mexico, so why is this location not listed?

Author's response on line 191: This was confusing because the original text did not properly describe the green markers. As described above, green markers are not coincident. Only the red markers are coincident in space and time, so we only describe the locations of the red markers.

Referee's comment on line 198: In Figure 5d there are many red markers depicting transit times >6 days, but here you claim domination by transit times of "two to five days".

Author's response on line 198: Thank you to the referee for catching this typo. Please note old Figure 5d is now Figure 7d in the revised manuscript.

**Author's change to manuscript for (old) line 198:**

We changed the text at line 248 in the revised manuscript from "two to five days earlier than" to "within seven days prior to".

Referee's comment on line 205: TX, OK and AR are in the "South Central" U.S.

Author's response and change to manuscript for (old) line 205: The referee is correct. At line 254 in the revised manuscript we have replaced "Central" with "South Central".

Referee's comment on line 216: Define MCC as mesoscale convective complex.

Author's response and change to manuscript for (old) line 216. This is a good point. The term is only used once, so at line 265 in the revised manuscript, we have replaced "MCC" with "mesoscale convective complex".

Referee's comment on line 227: Earlier you stated that back trajectories were initialized for every measurement of enhanced water vapor. Now, "Example" infers that this was done only for a subset. Did "all" (Line 228) of the back trajectories connect to OTs? Really? All?

Author's response on line 227: The referee has an excellent point here. Not all of the back trajectories connect to OTs, although a significant fraction of trajectories do connect to OTs (see Figure 10 and below in "changes to the manuscript"). Back trajectories were initialized at every 1-sec time stamp along the flight track in the "enhanced water vapor layer" (16-17.5 km altitude).

Author's changes to the manuscript in response to (old) line 227:

To characterize the fraction of trajectories that intersect OT, we introduce a new Figure 10 and new text at lines 280-282 in the Conclusions:

"For all the back trajectories in this analysis, the fraction that connect to OT within the previous seven days ranges from 30% to 70% (Figure 10)."

Also, we have deleted the old text "Example air parcel back-trajectories were initialized at the locations and time of enhanced water. All of the back-trajectories connect these air parcels to convective OT one to seven days prior to aircraft intercept."

and added the following text at lines 276-278 of the revised manuscript:

"Back trajectories initialized initialized at every 1-sec time stamp along the aircraft flight track at 16 to 17.5 km connect the sampled air parcels to convective OT within seven days prior to the flight."

Referee's comment on line 230: Figure 9b shows there were influences from storm systems in South Central Canada as well.

Author's response and changes for (old) line 230: The referee is correct. We have added to line 280 of the revised manuscript: "and South Central Canada."

Referee's comment on line 235: "leaving behind" should be "producing"

Author's response and change for (old) line 235: The referee is correct. We have changed the text at lines 286-287 of the revised manuscript to:

"... producing water vapor mixing ratios elevated 10 ppmv or more above background levels."

Referee's comment on line 238: "propel water" gives the wrong impression while "loft ice" is more accurate.

Author's response and changes to (old) line 238: The referee is correct. At line 289 in the revised manuscript we have changed "propel water" to "loft ice".

Referee's comment on line 239: "the enriched delta-D isotopic signature" needs a bit more explanation, including a mention of the isotope itself, HDO.

Author's response and changes to (old) line 239: The referee has a good point. We have modified the text starting at line 290 in the revised manuscript.

"Further evidence of ice is provided by water isotopologues. Evaporation and condensation are fractionating processes for isotopologues, especially HDO relative to  $H_2O$  (e.g., Craig, 1961; Dansgaard, 1964). Condensation preferentially concentrates the heavier HDO isotopologue, so lofted ice is relatively enriched in HDO/H2O compared to gas phase (e.g., Webster and Heymsfield, 2003, and references therein). Ice sublimation is supported by the enriched HDO/H2O isotopic signature observed by the ACE satellite over summertime North America (Randel et al., 2010)."

New references: Craig, H.: Isotopic Variations in Meteoric Waters, Science, 133, 1702-3, doi: 10.1126/science.133.3465.1702, 1961. Dansgaard, W.: Stable isotopes in precipitation, Tellus, 16, 436-68, 1964.

Referee's comment on line 246: "The water is almost certainly injected in the ice phase" needs support. I suggest you combined this paragraph with all or some of the previous paragraph that provides such support.

Author's response and changes to (old) line 246: Done. We have merged this paragraph (line 299 of revised manuscript) with the previous paragraph.

Referee's Comment on Line 252: "from a long (200 km) path through the atmosphere" also needs to include information about the vertical resolution. The qualitative statement "may be enhanced even more" is the basis for my argument that the MLS and aircraft results cannot be meaningfully integrated together in this paper.

Author's response and change to manuscript in response to (old) Line 252:

The simple answer about vertical resolution is ~3 km vertical resolution for MLS water in the lower stratosphere. We have replaced the text at (old) line 252 to the following new text at line 311 in the revised manuscript:

"Limb measurements from Aura MLS come from a ~200 km path through the atmosphere with ~3 km vertical resolution in the lower stratosphere (Livesey et al., 2016). The aircraft profiles of water vapor are very similar on ascent and descent profiling (Figure 6c), which allows us to estimate the horizontal length of these features as greater than 180 km, and a vertical thickness of ~0.5 km. This size is sufficiently large that the MLS retrieval is sensitive to enhanced water, as shown in Figure 6c."

Referee's Comment on Line 254: what percent of the 10 years of MLS observations show enhanced water?

Author's response and changes to paper for Line 254: our answer is in the revised paper starting at line 304 (for better flow):

"The fraction of Aura MLS observations at 100 hPa with H2O greater than 8 ppmv is only 0.9% for July-August 2013. In comparison, Schwartz et al. (2013) reports that, for the nine-year record 2004-2012, July and August had 1.4% and 3.2% of observations exceed 8 ppmv, respectively."

Referee's Comment on Line 261: Which "monsoon region"?

Author's response and change to manuscript for Line 261: Randel et al. (2015) addresses the North American Monsoon (NAM) region. We have changed the text in the revised lined 322 to: "NAM region."

Referee's Comment on old Figure 2 Caption: "ten year mean for 2004 through 2013."

Author's response on old Figure 2 Caption: We decided to compare 2013 with the previous nine-summer record, 2004 through 2012.

Author's changes to manuscript for old Figure 2 Caption (new Figure 4 caption): Old Figure 2 has been updated to new Figure 4 in the revised manuscript, and the caption changed to:

"Distribution of Aura MLS v4.2 100-hPa H2O over CONUS (blue shaded box in insert), corresponding to approximately 17 km altitude. The two histograms for July-August 2013 (blue asterisks and trace) and the previous nine-summer MLS record, July-August 2004 through 2012 (red circles and trace) indicates that 2013 was drier than average. The threshold for MLS-detected 'enhanced water vapor' (thick black vertical line) is set at 8 ppmv, same as Schwartz et al. (2013), to exclude the larger population of measurements at 6 to 8 ppmv water vapor that may have other sources."

Referee's Comment on old Figure 3 Caption: "Average Aura MLS 100-hPa"

Author's response and change to old Figure 3 Caption (new Figure 5): we changed the new Figure 5 Caption to "Two-month mean map of Aura MLS v4.2 100-hPa..."

Referee's Comment on old Figure 4: "[1] Colored markers are not "retrievals" from the aircraft, they are the actual aircraft measurements. [2] Why do multiple aircraft profiles produce only one profile convolved with the MLS averaging kernels? Without strongly magnifying this Figure I can't tell the difference between the MLS profiles and the convolved aircraft profile. I suggest omitting the convolved aircraft profile in each panel. It does shows that the averaging kernels smooth the aircraft profiles, but isn't that exactly what one would expect anyway? [3] Also, the black asterisks showing MLS retrieval locations on flight track maps are difficult to distinguish from black map lines. Perhaps larger gray symbols "x" or "+" would stand out more? [4] And the "line" mentioned in the caption, is this the horizontal line in each "Flight Altitude" (should be "Pressure") vs UTC hour panel indicating the time range of measurements shown in the profiles? [5] This Figure would be an ideal place to visually show (as a vertical line in each profile panel) the threshold for aircraft "enhanced" water vapor."

Author's response on old Figure 4: These are great points by the referee. We will implement changes as listed below.

Author's changes to old Figure 4 (Figure 6 in revised manuscript):

[1] We have changed the figure 6 caption from "water vapor retrievals from aircraft (color), aircraft with MLS averaging kernel (asterisk and lines)" to "Vertical profiles of *in situ* water vapor measurements from JLH Mark2 (dots) and MLS retrievals of water vapor (circles and lines)."

[2] As the referee recommended, we have omitted the convolved aircraft profile for improved clarity in Figure 6c (previously, multiple aircraft profiles were combined to produce one profile convolved with the MLS averaging kernels).

[3] We have changed the MLS retrieval symbols in Figure 6a to colored asterisks for clarity.[4] Yes, the MLS measurement times are signified by the colored horizontal lines in Figure 6b. The Pressure vs UTC hour panel is now altitude vs time (Figure 6b), properly labeled "Altitude", with new caption "MLS times (horizontal lines)".

[5] To address the referee's comment, we have added text to the Figure 6c caption: "Some measurements exceed the threshold for enhanced water vapor of 8 ppmv for Aura MLS (after Schwartz et al., 2013), and the campaign-wide mean plus 2 st. dev. for JLH Mark 2, 9.7 ppmv at 380-400 K and 6.6 ppmv at 400-420 K."

Referee's Comment on Figure 5: These comments apply to each of Figures 5-7: In panels (a), the captions claim the blue markers denote "enhanced water measurements" (which range below 5 ppmv) while in the panel (a) legends the blue markers are said to represent the "Enhanced H2O region", which must be something quite different from "enhanced water vapor" measurements. This distinction needs to be clarified in the paper by defining exactly what is meant by the terms "enhanced water measurements" and "enhanced H2O region".

Author's response and change to (old) Figures 5-7: We agree with the referee that this discussion is confusing, and have made changes to the manuscript. As described in lines 108-109, the threshold for 'enhanced water vapor' is mean + 2 sigma (from the combination of 23 aircraft flights) binned in two layers: 380-400 K and 400-420 K potential temperature. We observe these 'enhanced water vapor' events at 380-410 K (see Figure 1), so we call this layer the 'enhanced water region.'

New text added at lines 112-113 in the revised manuscript:

"We define the 'enhanced water region' as the layer of the overworld stratosphere where these events have been observed, 380-410 K potential temperature corresponding to 16-17.5 km altitude."

The blue points in (old) Figure 5 were confusing, so we removed them (see new Figures 7,8,9).

Authors' response to anonymous referee #2 on "Enhanced Stratospheric Water Vapor over the Summertime Continental United States and the Role of Overshooting Convection" by R. L. Herman et al., ACP-2016-1065.

We would like to thank the referee #2 for detailed comments (shown in quotations below). The individual points are addressed by the authors below:

**Referee's General Comment:**

Review of "Enhanced Stratospheric Water Vapor over the Summertime Continental United States and the Role of Overshooting Convection" by Herman et al.

"This paper presents direct airborne measurements of water injection into the lowermost stratosphere over the continental United States by convective overshooting tops and relates these to individual overshooting events through trajectory analysis. The study is generally well written, however, the overall result and conclusion is somewhat weak. I would recommend this paper for publication only after major revisions, for which I give suggestions below."

Author's response to general comment: we have addressed all of the referee's points below in the major and minor comments. Both the text and some figures have been modified in response to the referee's comments.

**Referee's Major comments:**

The observations themselves are not new and a number of previous studies have clearly indicated that overshooting convection may transport water ice into the stratosphere, where it evaporates and increases the stratospheric water vapor concentration. The novelty of this study is that it links observed water vapor enhancements to possible overshooting top events through trajectory analysis. This result, while new, is not very surprising and leaves the paper with a rather insignificant result. [1] The paper would benefit strongly from a discussion of the significance of this result and a much enhanced statistical analysis using their entire observational set. The authors indicated that they have many more observations during this campaign but chose to show only three examples. The authors might want to use their entire data set and increase their statistical analysis. [2] Their only statistical argument is at the end of the discussion, where they use only MLS data to state, that the impact is small. However, their own data (Figure 4) shows nicely, that MLS misses the highest concentrations due to its strong vertical averaging, which will heavily skew the result. Since the water vapor enhancements seem present on a very large scale, it would be good to see the entire data set for this campaign. The authors could then attempt to make a statistical analysis on how well they can relate these enhancements to OT events, what their temporal distribution may have been, and if there could be some preferred regions. [3] In the past water vapor instruments onboard the high altitude aircraft have shown significant disagreements. The authors state, that the other instruments show similar results. It would be good to actually show these, which would support the confidence in the observations themselves.

Authors' Response and Change to manuscript in response to Referee's Major Comment [1]: We agree with the referee that the paper would benefit strongly from a discussion of the significance and statistical analysis using the entire dataset. The significance is that the "enhanced" water measurements can be traced, for the first time, back to storm clusters with identified OT events in several common geographical areas. In the revised manuscript, three new figures (Figures 1, 2 and 3) characterize the distribution of stratospheric water vapor on all SEAC4RS flights. Out of 23 flights over the continental United States (CONUS), eleven showed enhanced water vapor. Details of these eleven flights are shown in the new Table 1, and discussed in the updated Section 2.1.

Authors' Response and Change to manuscript in response to Referee's Major Comment [2]: Aura MLS has a significant signal in  $H_2O$  from lower stratospheric enhanced water events. The threshold for MLS is 8 ppmv (Schwartz et al., 2013) to exclude points at 6-8 ppmv that may have other sources of water. The revised figure 6 (old figure 4) has been made clearer to demonstrate that the 'enhanced' water over the South Central U.S. (point 2 in this figure) is well above typical mixing ratios both for the aircraft data and the MLS satellite measurements.

**Authors' Response to Referee's Major Comment [3]:**

During SEAC4RS, the agreement between the two water vapor instruments on the ER-2 (JLH Mark2 and Harvard Water Vapor) is within +/-10% for stratospheric water. This is consistent with the AquaVIT laboratory intercomparison (Fahey et al., 2014). The Harvard data are not presented here because they will appear in a companion paper by J. Smith et al. In the past, other water vapor instruments had different biases, but we assert that those instruments would also show enhanced water well above their measured mean.

**Author's Change to manuscript in response to Referee's Major Comment [3]:**

New text added in revised manuscript at line 120-121:

"During SEAC4RS, the agreement between the two water vapor instruments on the ER-2 is within +/-10% for stratospheric water. This is consistent with the AquaVIT laboratory intercomparison (Fahey et al., 2014)."

**New Reference added to revised paper:**

Fahey, D. W., Gao, R.-S., Möhler, O., Saathoff, H., Schiller, C., Ebert, V., Krämer, M., Peter, T., Amarouche, N., Avallone, L. M., Bauer, R., Bozóki, Z., Christensen, L. E., Davis, S. M., Durry, G., Dyroff, C., Herman, R. L., Hunsmann, S., Khaykin, S., Mackrodt, P., Meyer, J., Smith, J. B., Spelten, N., Troy, R. F., Vömel, H., Wagner, S., and Weinhold, F. G.: The AquaVIT-1 Intercomparison of Atmospheric Water Vapor Measurement Techniques, Atmos. Meas. Tech., 7, 3177-3213, doi:10.5194/amt-7-3177-2014, 2014.

**Minor comments:**

**Referee's minor comment 1:**

"The manuscript should try to stick to one vertical coordinate and add other vertical coordinates only as additional information, e.g '90 hPa (370 K)'. Figure 4 uses pressure as vertical coordinate for consistency with MLS. Therefore, this could be the vertical coordinate system of choice. The profile figures may add approximate potential temperature as additional vertical axis for reference."

**Author's Response and Changes in Manuscript in response to comment 1:**

The referee has a good point here and the figures will be modified. To be consistent with previous literature on OT, the vertical coordinate of choice is altitude. We have changed Figure 6 (old Figure 4) vertical coordinates to "Altitude" (left axis) and "Approx. Pot. Temperature" (right axis). MLS Figures 4, 5, 6 (old Figures 1, 2 and 3) use the standard MLS 100-hPa product but we will modify each caption to "100 hPa (approximately 17 km altitude)..." In the Conclusions, we replaced "between 160 and 80 hPa" with "16 to 17.5 km altitude." (note, "160 hPa" was a typo and should have read "115 hPa").

**Referee's minor comment 2:**

"Most data shown in Figure 4 repeat between panels a-c. This figure could be combined into one panel with MLS data color coded roughly following the aircraft data."

Authors' Response and Change to manuscript in response to minor comment 2: Figure 6 (Old Figure 4) has been remade, with panels combined, and easier to read.

**Referee's minor comment 3:**

The use of green dots in Figures 5-7 is confusing. Panels c seem to indicate coincidences with relaxed conditions, whereas panels b seem to indicate all overshooting top events in the given time frame to show convective regions. This should be clarified.

**Author's response to minor comment 3:**

We agree with the referee that this was presented in a confusing manner. The description in the original text doesn't quite match what's on the figure. In panels b and c, the red symbols are where there were coincidences for the specific example trajectories plotted in light blue. Likewise, in panels b and c, the green symbols show all overshooting convective locations within +/- 3 hours of the red points, **not** related to where any of the trajectories went.

**Author's change to manuscript in response to minor comment 3:**

We removed "nearly coincident" from the manuscript since the only coincidence for green markers is in time. That doesn't really qualify as nearly coincident. The reason to show all the green symbols is to give an indication of how robust the coincidences are. For instance, the mass of green points north of Texas on 8 Aug 2013 indicates there were a lot of overshoots there and the coincidences should be robust for the trajectories that went through that region. The two coincidences in Arizona are among only a small cluster of overshoots and so this is not as robust a coincidence.

**Referee's minor comment 4:**

There are several references to "stratospheric background levels". [1] How where these background levels defined for this purpose? Are the profiles west of the Rocky Mountains considered "background" or did the authors use something else to define what the background is for this purpose? If they used the West Coast profiles, then they should briefly discuss the meteorology and exclude that these are more typical high latitude profiles. Could it be that the "background" is not as low as the authors assume? [2] There is obviously a large uncertainty in the detection and assignment of OT events. It would be good if the authors discussed how this uncertainty impacts their identification of possibly source events. [3] What is the lifetime of a typical overshooting top? [4] How many are likely to be missed by the OT detection algorithm? [5] Especially on the events that are closer to the observations, can the authors identify individual events that are best candidates?

Authors' Responses and Changes to manuscript in response to minor comment 4 [1]: The mean water vapor for all flights is 5.0 +/- 0.8 ppmv at 400 K

Author's response to comment 4 [5]: No, we cannot identify individual OT events for each back trajectory. There is considerable uncertainty in the detection and assignment of OT events, but the ensemble of trajectories repeatedly show coincidences with OT in specific storm clusters, especially for the 27 August 2013 case.

Author's changes to text in response to comment 4 [5]:

New text added in Section 3:

"Given uncertainties in back trajectories, GOES under-sampling, and that many OTs can be located in close proximity to one another, we are not able to make a direct connection between an individual OT and a stratospheric water vapor plume observed a day or more later. Rather, our analysis identifies a cluster of storms that are the best candidates for generating ice that sublimates into enhanced water vapor plumes sampled by the ER-2."

Referee's Comment on Lines 118-119: better: ' ...was drier than the 10 year MLS record ...' Authors' Response and change to manuscript: We changed the old lines 118-119 to reworded lines 135-137 in the revised manuscript (note that old Figure 2 is now Figure 4): "The histogram of Aura MLS water vapor in Figure 4 indicates that the July-August 2013 CONUS lower stratosphere was drier than the previous nine-summer MLS record (2004 to 2012)."

Referee's Comment on Lines 129-130: better '...the storm systems from which they may have originated, it is necessary...'

Authors' Response and change to manuscript: We agree with the referee, and changed the old lines 129-130 to the suggested text (lines 149-150 in the revised manuscript): '...the storm systems from which they may have originated, it is necessary...'

**1 Title**

2

**3 Enhanced Stratospheric Water Vapor over the Summertime**

**4 Continental United States and the Role of Overshooting Convection**

- 5 Robert L. Herman1, Eric A. Ray2, Karen H. Rosenlof2, Kristopher M. Bedka3, Michael J.
- 6 Schwartz1, William G. Read1, Robert F. Troy1, Keith Chin1, Lance E. Christensen1, Dejian Fu1,
- 7 Robert A. Stachnik1, T. Paul Bui4, Jonathan M. Dean-Day5
- 8 1Jet Propulsion Laboratory, California Institute of Technology, Pasadena, California, USA.
- 9 2National Oceanic and Atmospheric Administration (NOAA) Earth System Research Laboratory (ESRL) Chemical
- 10 Sciences Division, Boulder, Colorado, USA.
- 3NASA Langley Research Center, Hampton, Virginia, USA.
- 12 4NASA Ames Research Center, Moffett Field, California, USA.

[revised manuscript text omitted]
|-------------|-------------------|----------------|---------------|-------------------|----------------|---------------|
|             | above 400 K       | above 400 K    | above 400 K   | 380-400K          | 380-400K       | 380-400K      |
| 8-Aug-2013  | 10.1              | 401.2          | 17.29         | 11.2              | 385.7          | 17.10         |
| 12-Aug-2013 | 8.0               | 400.1          | 17.08         | 13.2              | 388.1          | 16.86         |
| 14-Aug-2013 | 7.7               | 402.2          | 17.38         | 10.7              | 387.4          | 16.75         |
| 16-Aug-2013 | 7.0               | 400.2          | 17.14         | 12.2              | 387.3          | 16.82         |
| 27-Aug-2013 | 15.3              | 402.8          | 17.32         | 17.7              | 380.8          | 16.12         |
| 30-Aug-2013 | 9.2               | 400.2          | 17.27         | 12.0              | 390.0          | 16.81         |
| 2-Sep-2013  | 8.0               | 400.3          | 17.07         | 13.0              | 380.3          | 16.28         |
| 4-Sep-2013  | 6.3               | 405.0          | 17.57         | 10.8              | 380.2          | 16.32         |
| 6-Sep-2013  | 6.8               | 400.1          | 17.12         | 15.6              | 381.0          | 16.32         |
| 11-Sep-2013 | 7.7               | 400.2          | 17.13         | 10.2              | 381.0          | 16.22         |
| 13-Sep-2013 | 6.9               | 401.8          | 17.55         | 9.2               | 382.4          | 16.41         |

503 504

\* SEAC4RS = Studies of Emissions and Atmospheric Composition, Clouds and Climate Coupling by Regional
 Surveys

505 506

| 507 | FIGURE | CAPTIONS |
|-----|--------|----------|

508

**Figure 1.** JLH Mark2 stratospheric water vapor profiles from 23 aircraft flights during SEAC4RS. This altitude

[revised manuscript text omitted]

---

## Editor Decision (ED1)

**A few comments on "Enhanced stratospheric water vapor..."**

There was also a comparison of  hygrometers during the NASA MACPEX mission; this study might also be helpful here:

Rollins, A. W., et al. (2014), Evaluation of UT/LS hygrometer accuracy by intercomparison during the NASA MACPEX mission, J. Geophys. Res. Atmos., 119, 1915–1935, doi:10.1002/2013JD020817.

Tissier and Legras also discuss the accurracy of the OT altitude and obtain a somewhat different results than reported here (l. 169)

Tissier, A.-S. and Legras, B.: Convective sources of trajectories traversing the tropical tropopause layer, Atmos. Chem. Phys., 16, 3383-3398, doi:10.5194/acp-16-3383-2016, 2016.

The authors argue on the efficiency of mixing by breaking gavity waves (l. 188 and l. 296). Please consider this citation from a review of another paper (review by T. Dunkerton):
*"..vertical diffusion to breaking gravity waves, it is well-known that such waves (if undergoing local convective instability in their phase of overturning) are not effective in mixing heat and constituents vertically (Coy et al., 1988 JAS). Inertia-gravity waves may undergo shear instability at large amplitude, altering this result possibly in a significant way (Dunkerton, 1985 JAS, et seq.). "*

l.257: I find this sentence confusing, please consider to rephrase.

---

## Author Response (AR2)

Response to the co-editor on SEAC4RS OT manuscript acp-2016-1065.

Dear co-editor and staff,

We thank the co-editor and two reviewers for very helpful comments and constructive revision requests. We have responded to all comments and minor revision requests as listed below.

**Response to co-editor:**

Editor's comment 1: There was also a comparison of hygrometers during the NASA MACPEX mission; this study might also be helpful here: Rollins, A. W., et al. (2014).

**Author's response to comment 1 and change in manuscript:**

The Rollins et al (2014) paper puts these measurements into perspective, namely level of agreement between JLH and Harvard Water Vapor, so we have cited this paper in the text at line 121-122:

"This is consistent with the AquaVIT laboratory intercomparison (Fahey et al., 2014) and other aircraft field missions (e.g., Rollins et al., 2014)."

New reference:

Rollins, A., Thornberry, T., Gao, R. S., Smith, J. B., Sayres, D. S., Sargent, M. R., Schiller, C., Krämer, M., Spelten, N., Hurst, D. F., Jordan, A. F., Hall, E. G., Vömel, H., Diskin, G. S., Podolske, J. R., Christensen, L. E., Rosenlof, K. H., Jensen, E. J., and Fa- hey, D. W.: Evaluation of UT/LS hygrometer accuracy by inter- comparison during the NASA MACPEX mission, J. Geophys. Res.-Atmos., 119, doi:10.1002/2013JD020817, 2014.

Editor's comment 2: Tissier and Legras also discuss the accuracy of the OT altitude and obtain a somewhat different result than reported here (1. 169)

**Author's response to comment 2:**

We assume that the editor is referring to the following comment made in Tissier and Legras: "We use the CLAUS data set (Hodges et al., 2000), which provides global 3-hourly maps of brightness temperature at 30km resolution, combined with ERA-Interim data (Dee et al., 2011) to determine the pressure of the top of the convective clouds.....This method is, however, limited by the inability to distinguish overlaying cirrus from convective tops and is known to underestimate the altitude of the deep convective clouds by about 1 km (Sherwood et al., 2004; Minnis et al., 2008)." The editor is correct in that, in situations with above anvil cirrus plumes (i.e. "overlaying cirrus from convective tops") described by Setvak et al in numerous papers and Homeyer (JAS, 2014) and et al. (JAS, 2017), the IR brightness temperature (BT) in these plumes is anomalously warm because the cirrus adjusts to the stratospheric temperature warmer than the tropospheric primary anvil cloud. If one matches the warm IR BT to a sounding and only considers to height assign the cloud to the troposphere, then yes there will be a low bias in cirrus plume heights. Above anvil cirrus only occurs with a small subset of all overshooting convection though. We are looking at convective cores that generate the plumes due to favorable wind shear inducing gravity wave breaking. Cores (i.e. OTs) are significantly colder than the surrounding anvil, and the Griffin et al approach is based on this premise. So, the editor's comment about a discrepancy in height assignment errors is not an "apples to apples" comparison of phenomena and thus is not relevant to our paper or results.

Editor's comment 3: The authors argue on the efficiency of mixing by breaking gravity waves (1. 188 and 1. 296). Please consider this citation from a review of another paper (review by T. Dunkerton):

"..vertical diffusion to breaking gravity waves, it is well-known that such waves (if undergoing local convective instability in their phase of overturning) are not effective in mixing heat and constituents vertically (Coy et al., 1988 JAS). Inertiagravity waves may undergo shear instability at large amplitude, altering this result possibly in a significant way (Dunkerton, 1985 JAS, et seq.). "

**Author's response to comment 3:**

In our manuscript (line 188), we cite recent references in the literature: "turbulent processes such as gravity wave breaking mix tropospheric and stratospheric air (e.g., Mullendore et al., 2009, 2005; Wang 2003; Homeyer et al. 2017)." With regards to the Dunkerton comment, Homeyer et al (2017) and Wang (2003) clearly show water vapor enhancement in the stratosphere due to these waves. Perhaps Dunkerton's findings seemed robust at the time, but advancements in satellite imagery and modeling capabilities have since highlighted the importance of these waves for generating the detrainment and irreversible mixing of ice (and thus water vapor via sublimation) in the stratosphere.

Editor's comment 4: 1.257: I find this sentence confusing, please consider to rephrase.

"This case is an example of the classic North American Monsoon circulation with a moisture source over the Sierra Madre Occidental. Anticyclonic transport carried the moisture north from Mexico, and counter-clockwise around the high pressure (Figure 8b)."

**Author's response to comment 4 and change in manuscript:**

The editor is correct that this wording is confusing. Figure 8b shows the dominant anticyclonic transport of the North American Monsoon. We have reworded and merged the two sentences as follows:

"This case is an example of the classic North American Monsoon circulation with a moisture source over the Sierra Madre Occidental (Adams and Comrie, 1997), in which air parcels are transported from OT in Mexico, around the anticyclone, to the CONUS (Figure 8b)."

**New reference:**

Adams, D. K., and A. C. Comrie, 1997: The North American Monsoon. *Bull. Amer. Meteor. Soc.*, 78, 2197–2213, doi:http://dx.doi.org/10.1175/JCLI4071.1.
* * *
**Response to Report #1**

This is our response to the minor comments from referee #1:

**Referee's Comment 1:**

First, I would find it a lot more informative to combine Figures 2 and 3 to visually compare the mixing ratio distributions below and above 400K. The y-axis would need to be changed to "fraction of observations" by dividing each histogram by the number of observations it contains. In other words, the sum of fractions within each histogram would be unity.

**Author's response to comment 1 and changes to manuscript:**

We agree with referee #1 that it would be good to combine Figures 2 and 3. The new Figure 2 in the revised manuscript combines these two histograms, y-axis changed to "fraction of observations" and data normalized to the number of observations.

**Referee's Comment 2:**

Second, with the introduction of Figure 10 there now exists a real contradiction in the paper. Figure 10 implies that, on average, about 50% of the air masses sampled

between 370 and 420K during the three studied flights were connected to convective overshooting tops. This is in stark contrast to the 5% of air masses sampled between 370 and 420K that had "enhanced water". I think the problem lies in the very high (9.7 ppmv) threshold for enhanced water in the 380-400K layer. From Figure 2 the distribution of mixing ratios is skewed to higher values and is quite non-Gaussian. Thus, the choice of mean + 2\*stddev as the threshold yields a very high value. I would be tempted to try a different a different statistic, perhaps involving the median of the highly skewed distribution, to calculate a threshold. I realize this threshold is subjectively determined, but the contradiction the current threshold creates leaves the reader somewhat perplexed: Why aren't 50% of the measurements considered "enhanced water" when 50% of the air masses were connected to OTs? Don't be concerned that the enhanced water fraction from the JLH measurements is much higher than that from MLS because the three SEAC4RS flights were targeting air masses influenced by convection. And to avoid any mis-interpretation of 50% enhanced water and Figure 10 (as supporting Anderson et al. 2012) I would make sure to clearly state that the three SEAC4RS flights targeted air masses influenced by convection.

**Author's response to Comment 2:**

We appreciate this excellent comment by the reviewer.

In this manuscript, we **conservatively** defined "enhanced water" as mean + 2\*stddev to exclude the larger population of measurements at 6 to 9.7 ppmv water vapor that may have other sources. Figures 1 and 2 show the statistics of the entire ensemble of 23 aircraft flights, some of which did not target air masses influenced by convection. Thus, the Figure 2 distribution shows only a small fraction of CONUS air parcels with "enhanced water." Figure 10 shows only three flights, the case studies, with 50% "enhanced water."

This apparent contradiction is resolved by a new statement to be added that the three SEAC4RS flights in Figure 10 (now renumbered to Figure 9) deliberately targeted air masses influenced by convection. Therefore the fraction of mass masses influenced by convective OT is higher than for the entire CONUS.

Author's change in manuscript in response to Comment 2:

The new text at lines 281 to 285 is:

"For all the back trajectories in the three case studies, the fraction that connect to OT within the previous seven days ranges from 30% to 70% (Figure 9). The three aircraft flight dates analyzed in Figure 9 have a higher fraction of enhanced water than the other flights. These three flights deliberately targeted air masses

influenced by convection. For the CONUS in general, the fraction of air parcels at 370-420 K influenced by OT is much smaller."

Response to Report #2

\_\_\_\_\_

Referee #2 says that "the authors have significantly improved the manuscript. Together with the changes made in response to the first reviewer, the authors have addressed all my concerns."

We thank referee #2, and have completed all Technical corrections recommended by referee #2:

Line 23: Better write: '... satellite infrared window-channel brightnesstemperature gradients ...' (note the hyphens). Done.

Line 111: 'H2O is 6.7' (delete for) Done.

Line 111: 'Thus, ...' (added comma) Done.

Line 152: 'Data for' (not From) Done.

Line 197: which should it be: 'storm-updraft tracks' or 'storm updraft-tracks'? Changed to "tracks of storm updrafts"

Line 212: delete 'also' Done.

Line 274: 'In this paper, ...' Done.

Line 276: duplicate 'initialized' Done.

Figure 4: Please remove grey background.

Done.

**1 Title**

- 2
- 3 Enhanced Stratospheric Water Vapor over the Summertime

**4 Continental United States and the Role of Overshooting Convection**

- 5 Robert L. Herman1, Eric A. Ray2, Karen H. Rosenlof2, Kristopher M. Bedka3, Michael J.
- 6 Schwartz1, William G. Read1, Robert F. Troy1, Keith Chin1, Lance E. Christensen1, Dejian Fu1,
- 7 Robert A. Stachnik1, T. Paul Bui4, Jonathan M. Dean-Day5
- 8 1Jet Propulsion Laboratory, California Institute of Technology, Pasadena, California, USA.
- 9 2National Oceanic and Atmospheric Administration (NOAA) Earth System Research Laboratory (ESRL) Chemical
- 10 Sciences Division, Boulder, Colorado, USA.
- 11 3NASA Langley Research Center, Hampton, Virginia, USA.
- 12 4NASA Ames Research Center, Moffett Field, California, USA.

[revised manuscript text omitted]
|-------------|-------------------|----------------|---------------|-------------------|----------------|---------------|
|             | above 400 K       | above 400 K    | above 400 K   | 380-400K          | 380-400K       | 380-400K      |
| 8-Aug-2013  | 10.1              | 401.2          | 17.29         | 11.2              | 385.7          | 17.10         |
| 12-Aug-2013 | 8.0               | 400.1          | 17.08         | 13.2              | 388.1          | 16.86         |
| 14-Aug-2013 | 7.7               | 402.2          | 17.38         | 10.7              | 387.4          | 16.75         |
| 16-Aug-2013 | 7.0               | 400.2          | 17.14         | 12.2              | 387.3          | 16.82         |
| 27-Aug-2013 | 15.3              | 402.8          | 17.32         | 17.7              | 380.8          | 16.12         |
| 30-Aug-2013 | 9.2               | 400.2          | 17.27         | 12.0              | 390.0          | 16.81         |
| 2-Sep-2013  | 8.0               | 400.3          | 17.07         | 13.0              | 380.3          | 16.28         |
| 4-Sep-2013  | 6.3               | 405.0          | 17.57         | 10.8              | 380.2          | 16.32         |
| 6-Sep-2013  | 6.8               | 400.1          | 17.12         | 15.6              | 381.0          | 16.32         |
| 11-Sep-2013 | 7.7               | 400.2          | 17.13         | 10.2              | 381.0          | 16.22         |
| 13-Sep-2013 | 6.9               | 401.8          | 17.55         | 9.2               | 382.4          | 16.41         |

[revised manuscript text omitted]

643 Figure 1. JLH Mark2 stratospheric water vapor profiles from 23 aircraft flights during SEAC4RS. This altitude

range includes the overworld stratosphere (potential temperature greater than 380 K) and lowermost stratosphere
(tropopause to 380 K). The majority of observations have mixing ratios less than 10 ppmv in the lowermost

646 stratosphere and less than 6 ppmv in the overworld stratosphere. Enhanced water measurements are the extreme

647 outliers with high water mixing ratios, with a threshold value of mean plus two standard deviations.